# GameDevBench: Evaluating Agentic Capabilities Through Game Development

**Wayne Chi** [1]  **Yixiong Fang** [1]  **Arnav Yayavaram** [1]  **Siddharth Yayavaram** [1]  **Seth Karten** [2]  **Qiuhong Anna Wei** [1]
**Runkun Chen** [1]  **Alexander Wang** [1]  **Valerie Chen** [1]  **Ameet Talwalkar** [1]  **Chris Donahue** [1]

## Abstract

Despite rapid progress on coding agents, progress on their multimodal counterparts has lagged behind. A key challenge is the scarcity of evaluation testbeds that combine the complexity of software development with the need for deep multimodal understanding. In game development, agents must navigate large, dense codebases while manipulating intrinsically multimodal assets such as shaders, sprites, and animations within a visual game scene. We present `GameDevBench`, the first benchmark for evaluating agents on game development tasks. `GameDevBench` consists of 333 tasks derived from web and video tutorials. Tasks require significant multimodal understanding and are complex—the average solution requires over three times the lines of code and file changes compared to prior software development benchmarks. Agents struggle with game development, with the best agent and method solving only 53.8% of tasks. We find a strong correlation between perceived task difficulty and multimodal complexity, with average success rate dropping from 51.4% on gameplay-oriented tasks to 33.0% on 2D graphics tasks. To improve multimodal capability, we introduce two simple image and video-based feedback mechanisms for agents. Despite their simplicity, these methods consistently improve performance, increasing `GPT-5.4`'s performance from 41.1% to 52.0% when given visual feedback. We release our code at https://github.com/waynchi/gamedevbench

## 1. Introduction

Progress on multimodal language model (LM) agents has lagged behind that of their unimodal counterparts (Yang et al., 2024b; Jimenez et al., 2024; Zhou et al., 2024; Koh et al., 2024). Agentic game development—despite its inherent multi-modality, increasing public interest, and a rich history combining artificial intelligence and games (Vinyals et al., 2019; Schrittwieser et al., 2019; Silver et al., 2018; 2016; Jagli et al., 2024; Filipović, 2023; Yakan, 2022)—has largely been overlooked by the research community. Most prior works focus on specific goals within game development such as next frame prediction (Valevski et al., 2024; Oh et al., 2015), which replaces the graphics engine, procedural content generation (Summerville et al., 2018; Shaker et al., 2016), which replaces asset creation, or game playing agents (Vinyals et al., 2019; Silver et al., 2016), which replaces the non-player characters (NPCs) and opponents. There has been little to no research on agentic use for general game development (i.e., developing games within a game engine), most likely because it seemed inconceivable until recently. As LM agent capabilities continue to improve, it seems natural to ask: can agents develop video games?

Game development combines many desirable characteristics for a challenging benchmark in a modern agentic domain. First, tasks are **complex and context-rich** with projects often spanning large amounts of files, assets, and folders akin to that of traditional software development (Yang et al., 2024b). Second, tasks are inherently **multimodal**, requiring visual understanding of both static elements (e.g., map or scene layouts) and temporal dynamics (e.g., animations or movement) to accurately assess project state. Lastly, task solutions are **deterministically verifiable** through code which alleviates the need for approaches such as LLM-as-a-Judge (Zheng et al., 2023) which are often subject to biases (Wang et al., 2023; Koo et al., 2024). For example, it is possible to verify that the correct animation was used by checking animation states at each frame. This combination of features makes game development an ideal environment to evaluate complex, multi-modal agentic capabilities.

In this work, we study an agent's ability to solve complex game development tasks for a modern game engine. To our knowledge, this is the first work evaluating this capability. Game development typically involves creating and editing artifacts such as sprite sheet animations, collision shapes, game logic scripts, and scene layouts in a GUI (Graphical User Interface) called the game editor. A game engine

[1]Carnegie Mellon University, Pittsburgh, PA, USA [2]Princeton University, Princeton, NJ, USA. Correspondence to: Wayne Chi <waynechi@andrew.cmu.edu>.

*Proceedings of the $43^{rd}$ International Conference on Machine Learning*, Seoul, South Korea. PMLR 306, 2026. Copyright 2026 by the author(s).

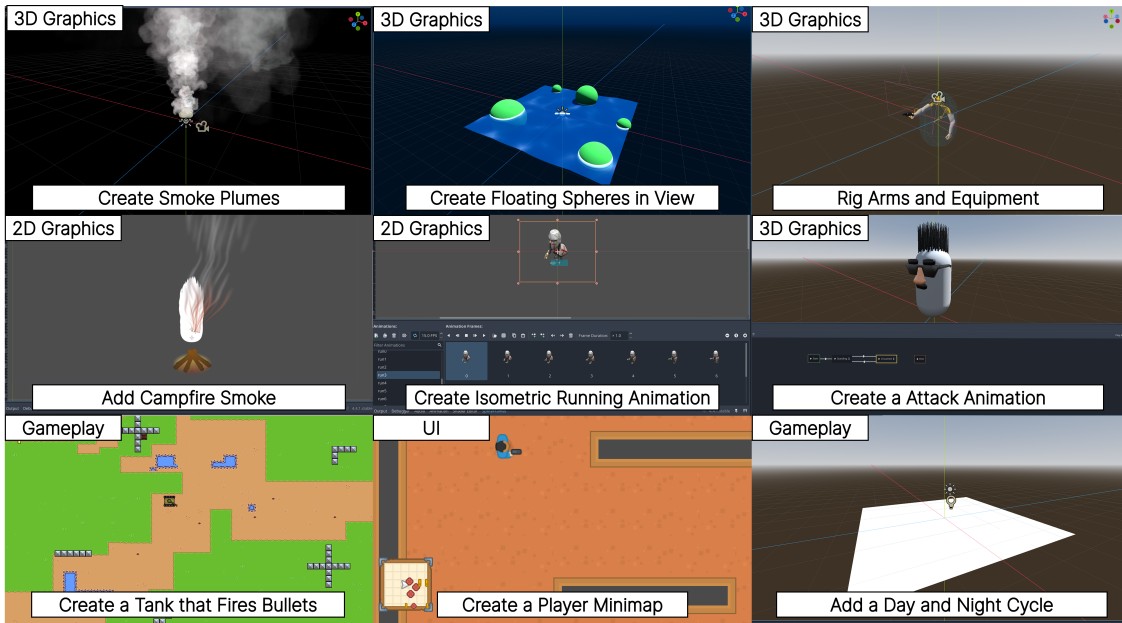

*Figure 1.* We present `GameDevBench`, a benchmark for evaluating an agent's ability to solve complex and multimodal game development tasks in a modern game engine.

then processes these artifacts into a runnable game build. Common examples of game engines include Unity, Unreal Engine, and Godot, each of which provides both an editor and an engine. Game development tasks are deceptively complex. For example, the "simple" task of creating an Italian plumber for a platformer game would require creating animations for various states such as idling, jumping, or running, setting up a collider to allow for jumping on enemies such as turtles, writing scripts to allow for control, adding sound effects for actions, and more.

We focus our work on the Godot environment for several reasons. First, Godot is fully open sourced under the MIT license which makes it easy to extend and release alongside the benchmark. Second, Godot is an increasingly popular game development engine, with 770 and 1185 releases on Steam in 2024 and 2025. Third, Godot's environment strongly resembles Unity, which is by far the most popular game development engine. Lastly, Godot projects (not including assets such as images) can be represented in code which makes it simple to extend existing LLM agent capabilities without having to construct specific tool-use APIs.

We present `GameDevBench`, the first benchmark for evaluating an agent's ability to solve game development tasks. Tasks are created by analyzing and processing Godot YouTube and web tutorials. These tutorials span a wide range of topics such as 2D sprite animations, character controllers (i.e., character movement), colliders and platforms, shader usage, particle effects, among others. This ensures that tasks are not only diverse, but also align with common game development needs. Tasks are incredibly complex and

content-rich. Not only do they require a deep understanding of various file types and assets (e.g., images), tasks on average require more than *three times* the number of lines of code changes compared to SWE-Bench (Yang et al., 2024b). For each task, agents are given a project folder with code and various assets, as well as an instruction as is standard in software benchmarks (Yang et al., 2024b). Task success is evaluated using tests built within Godot's scripting framework. This allows us to deterministically test for features such as physics or polygonal shapes. Additionally, each task comes with a verified reference solution. All code and task project files for `GameDevBench` are released publicly.

We found that while agents are increasingly capable, they still struggle with the majority of game development tasks. With additional multimodal support, the best agent succeeds at only 53.8% of the tasks. In particular, models perform significantly worse when the tasks require increased multimodal understanding. For example, agents perform nearly fifty percent as well on gameplay-oriented tasks compared to 2D graphics tasks (51.4% vs 33.0% averaged across all evaluated agents).

To improve agent multimodal capabilities, we propose two methods that provide agents with multi-modal feedback when solving a task. One method provides a screenshot view of the editor's current state via a Model Context Protocol (MCP) server (Anthropic, 2024) while another records a video of the game scene. Despite their simplicity, we found that both methods are effective empirically, increasing agent performance across almost all models.

## 2. Benchmark Construction

`GameDevBench` consists of game-development tasks distilled from online tutorials (e.g., "add a walking animation using the given spritesheet"). GitHub repositories are a rich source of data, but can be noisy and poorly documented. Additionally, unlike prior work (Jimenez et al., 2024) on benchmarking general software development, there are no obvious popular open source game repositories to choose from. The game development community, however, has created an abundance of online tutorials—many of which come with solution repositories—that guide developers through common development use cases. We use a multi-step pipeline to construct game development tasks using these tutorials.

### 2.1. Stage 1: Data Preparation

Game development tutorials primarily come in either text or video formats. For all tutorials, we search for and filter to only include Godot 4 tutorials that include a corresponding GitHub repository with permissive, open-source licenses.

**Video.** We source our video tutorials from YouTube. To convert video into text, we use a popular YouTube transcription API[1] to extract the text transcript from each video. To search for a matching GitHub repository, we parse the video description for any GitHub links. The final result is a folder for each tutorial containing the transcript and the corresponding GitHub repository. We process 102 video tutorials which were selected based on view count. Each tutorial averages 29 minutes of content. In the end, we use 57 tutorials as not all tutorials are usable due to non-functional repositories or mislabeled Godot versioning.

**Web.** For web text tutorials, we source from "Godot Recipes by KidsCanCode" (KidsCanCode), which is listed on the community resources page in Godot's official documentation. We scrape the webpages using a Python script with the goal of mirroring the structure of processed video tutorial folders. The end result is 99 tutorial folders, each of which contains the tutorial text content, a media directory of visual data downloaded from the webpage, a GitHub repository, as well as a metadata JSON containing information such as the tutorial URL. Finally, we ask an LLM to sort tutorials based on suitability for task creation and use the top 31 tutorials for subsequent task construction.

### 2.2. Stage 2: Automatic Task Construction

Given the tutorial folder, the agent is asked to create tasks where 1) instructions adhere to the tutorial, 2) task files are created directly based on existing files in the repository, and 3) unit tests must only test for features explicitly requested in the instructions. Access to the solution repos-

itory is crucial as it allows the agent to create tasks that it would not normally have the capability to solve or create. At the agent's discretion, each tutorial is split into multiple tasks to capture more well-defined skills. For example, the agent could decompose a platformer tutorial into tasks on character animation, controls and colliders, and tilemap construction. We use the `Codex` Agent with the GPT-5 family of models to construct tasks from each tutorial. `Codex` was chosen primarily due to its API limits and availability at the time; we did not notice any significant differences between agents such as `Claude Code`. We create 202 initial tasks with an average of 1.3 tasks per tutorial. The full prompt can be found in Appendix A.

### 2.3. Stage 3: Task Refinement

After stage 2, we found the majority of tasks to be sensible at a high level (i.e., task instructions were reasonable and matched the tutorial). However, similar to prior work (Chi et al., 2025), the agent was not able to perfectly create tasks and tests. We conducted a preliminary study on a small subset of 41 tasks where human annotators reviewed tasks and documented any issues observed. The study found that 43% of tasks were issue free, 50% of tasks had issues that required minor updates such as scenes being off-camera, tests asserting for non-existent instructions, or accidental references to other portions of the tutorial, and 7% of tasks contained major issues that made them difficult to fix. Since most of the errors were minor and easily caught, we employed a hybrid process to refine tasks. Based on the preliminary study, we constructed prompts and checklists to catch the most common mistakes. We then employed an agent to automatically verify and fix those mistakes based on the checklists. We re-use this prompt (Appendix B) when processing all future tutorials and tasks.

### 2.4. Stage 4: Human Annotation.

Lastly, 8 human annotators, 5 of whom have prior game development experience, reviewed all tasks. Annotation served three goals. First is to ensure that tasks are verified for correctness and resolvability as is common practice (Yang et al., 2024b; Chi et al., 2025). Annotators are instructed to look for and fix any ambiguous instructions, conflicting instructions, and overly strict tests. Additionally, annotators are asked to mark and remove any tasks that had any other issues. Second, we ask annotators to create variations of existing tasks similar to prior work (Zhou et al., 2024). An example would be two tasks that differ based on the requested animation used in a spritesheet (e.g, selecting the walking vs running animation frames). In total, we create 154 tasks and 179 task variants. Lastly, we asked annotators to annotate whether tasks were considered `easy` or `hard` based on the task's required multimodal understanding. In total, we create 131 `easy` and 202 `hard` tasks.

---

[1]https://github.com/jdepoix/youtube-transcript-api

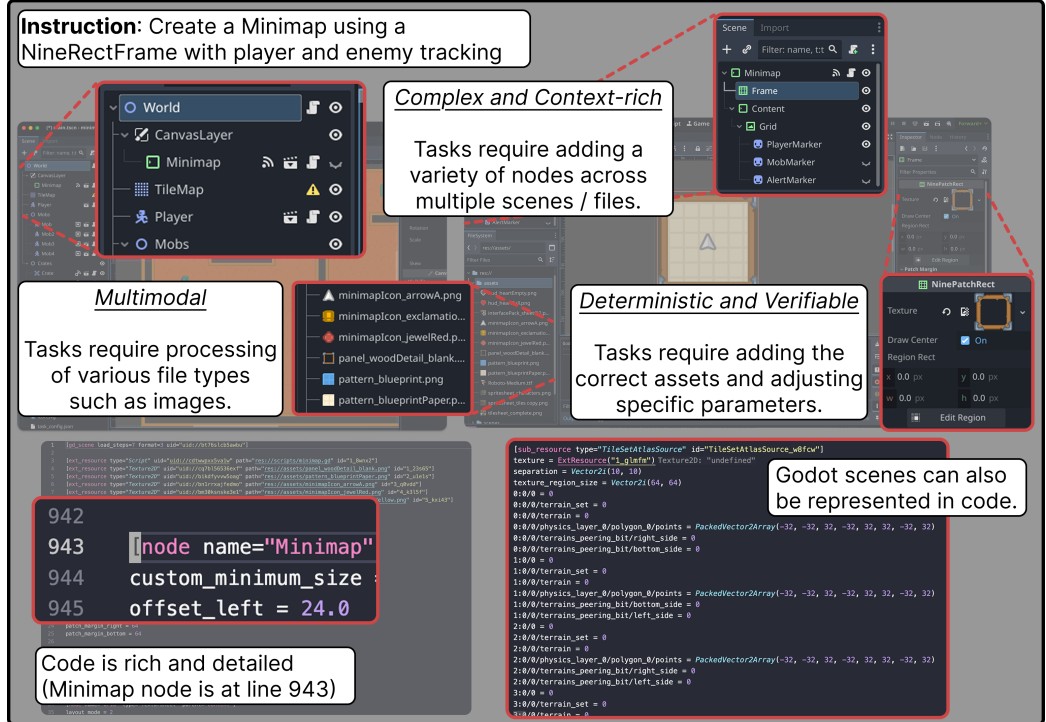

Figure 2. This is an example task from `GameDevBench` that requests for the creation of a UI minimap. Top is the visual GUI representation and highlighted points of interest. Bottom is the same scenes and files represented in code. Tasks can be solved via the editor or entirely through code although either method requires understanding multimodal assets. Game development tasks are complex and require editing dense files, identifying and visually understanding various assets, and navigating various nodes (game elements) and scenes (a collection of nodes).

Annotation instructions can be found in Appendix C.

## 3. GameDevBench

Game development sits at the intersection of creative expression and software development. As such, `GameDevBench` features a diverse set of tasks that are inherently multimodal, complex and context-rich.

### 3.1. Task Categories

To our knowledge, there is no existing taxonomy of game development tasks performed within a game editor or game engine. To better understand our task diversity and enable deeper analysis, we categorize each task along two axes.

**Categorization by skill set.** We induce a task categorization based on the underlying game development skills required by each task (Table 1). Specifically, we adopt a bottom-up categorization procedure: we first obtain fine-grained skill annotations for each task by asking `GPT-5-mini` to label each task. Labels are then abstracted into higher-level skill categories through a separate request to `GPT-5-mini`. These categories are subsequently reviewed and refined by game developers to ensure consistency and validity. This

process yields four skill categories: 2D graphics and animation, 3D graphics and animation, gameplay logic, and user interface.

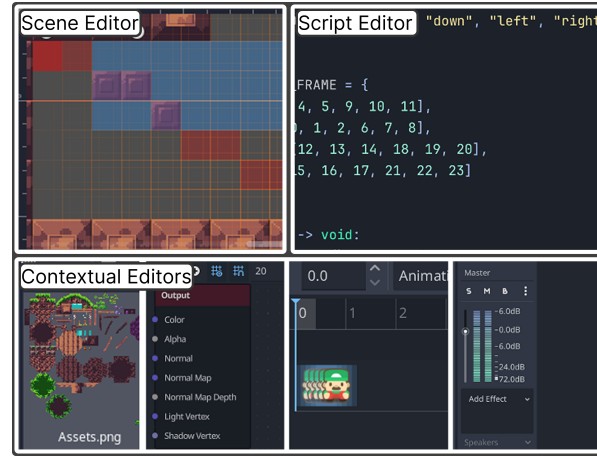

Figure 3. Types of editors in Godot. Top-left is the scene editor. Top-right is the script editor. The bottom contains various contextual editors. From left to right: tilemap, shader, animation, and audio editors. Contextual editors surface depending on use case. Typically, tasks that use contextual editors require deeper multi-modal understanding.

*Table 1.* Skill categories for Godot-related development tasks.

| Skill | Definition | Examples | % Tasks |
|---|---|---|---|
| 2D Graphics and Animation | Tasks involving the creation, rendering, and animation of two-dimensional visual content. | Sprite animation, TileMap setup, 2D shader effects | 33.3% |
| 3D Graphics and Animation | Tasks involving the construction, rendering, and animation of three-dimensional scenes. | Material tuning, Skeletal animation, Camera rigs | 26.7% |
| User Interface | Tasks concerning the design and implementation of interactive user interfaces. | HUD layout, Menu navigation, UI theming | 20.1% |
| Gameplay Logic | Tasks focused on implementing game rules and behaviors such as motion and collisions. | Enemy AI states, Signal-driven events, Collision detectors, Character controllers | 19.8% |

**Categorization by editor type.** Godot contains several different types of editors that users use to resolve various tasks. There are three main types of editors within Godot. The scene editor (Figure 3, top-left) allows the user to modify the game scene by constructing level maps or placing and editing objects. The script editor (Figure 3, top-right) is a built-in code editor. Contextual editors appear on the bottom panel depending on what the user is editing (Figure 3, bottom). For example, when editing an animation resource, the animation editor will appear. Some of the most common contextual editors include the animation, audio, shader, and tileset editors. We categorize each task in the benchmark by asking `GPT-5-mini` to determine the editors that a user would need to solve the task. While the agent may not directly interact with these editors, the type of editor a user would use provides a strong proxy for task categorization.

For simplicity, we assign each task one skill category and one editor type. We provide multiple examples with their skill category and editor type in Appendix E.

### 3.2. Features of `GameDevBench`

`GameDevBench` has a unique set of features which we describe as follows. We provide additional task statistics in Appendix F.

**Diverse file types across tasks.** Unlike agentic benchmarks in the software domain (Jimenez et al., 2024), `GameDevBench` requires that agents handle a wide variety of filetypes across various modalities (Figure 4, left). In fact, the vast majority of tasks (82.4%) contain additional assets such as images (.png), text fonts (.ttf), shaders (.gdshader), audio (.wav), and other asset resources (.tres). As such, `GameDevBench` inherently tests the multimodal capabilities of agents.

**Diverse task types.** While there are other domains (such as frontend development (Zhu et al., 2025; Si et al., 2024) or slide generation (Liang et al., 2025)) that intersect multimodality and code generation, most of these domains focus on tasks similar to user interface generation. On the other hand, `GameDevBench` features a diverse task set. Across

the 333 benchmark tasks, domains are distributed as follows: (33.3%) 2D Graphics and Animation, (26.7%) 3D Graphics and Animation, and (20.1%) User Interface, (19.8%) Gameplay Logic.

**Complex and context-rich solutions.** Similar to software tasks, `GameDevBench` solutions require multi-location edits that weave together multiple files. Our reference solutions average 4.7 files and 114.1 lines of code changed across 3.2 distinct filetypes (Figure 4, right). This is more than triple the number of lines of code and file changes required compared to SWE-Bench (Jimenez et al., 2024), suggesting substantial complexity to the tasks and corresponding solutions.

**Deterministic verification of multimodal solutions.** Evaluating multimodal solutions is inherently challenging and solutions are typically evaluated through metrics such as CLIP (Radford et al., 2021) or a Visual LLM-as-a-Judge (Yin et al., 2026). These methods are, however, either proxies to correctness or non-deterministic. `GameDevBench` instead uses Godot's testing framework which allows us to directly test game behavior. For example, we can check to see if objects are in view of the camera or if object colliders have interacted purely through unit tests. Thus, tests are repeatable and verifiable similar to software benchmarks while testing multimodal problems.

**Flexible solution methods.** While the tests are deterministic, the methods used to derive solutions are flexible. In this work, for simplicity, we evaluate agents that attempt to solve tasks through code generation alone. However, it would be equally feasible to solve each task directly in the editor with approaches more similar to computer use. Our test-based verification allows for direct comparison of different solution strategies.

**Continually Renewable.** While not unique to our benchmark, our pipeline is repeatable and thus the benchmark can be continuously renewed. Human validation is minor with each task taking under 10 minutes to validate.

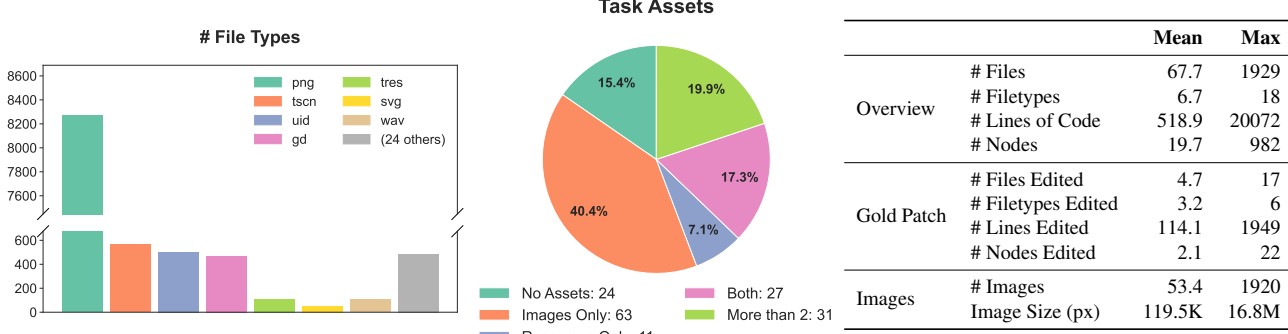

*Figure 4.* `GameDevBench` features a diverse amount of filetypes (31 different types, left). The vast majority of tasks contain either images, resources (e.g., Shaders), or multiple asset types (middle). Each task contains multiple scripts and scenes, both of which are context-rich and require a significant amount of tokens to process (right).

# 4. Evaluation

We evaluate various models and agentic harnesses on `GameDevBench`.

**Model Choices.** From the Claude family of models we evaluate `Claude Haiku 4.5` and `Claude Sonnet 4.5`. From the Gemini family we evaluate `Gemini 3 Flash` and `Gemini 3 Pro`. From the GPT family we evaluate `GPT-5.4 Mini` and `GPT-5.4`. For open weights models we evaluate `Qwen3.5-397B` and `Kimi K2.5`.

**Agent Harness Choices.** To allow agents access to both the project files and the Godot application itself, we focus on agentic harnesses that operate locally. We chose command-line interface (CLI) harnesses due to their ability to directly read code, image, and other asset files. We evaluate each model in its respective agentic harness—`claude-code` for Claude models, `gemini-cli` for Gemini models, and `codex` for ChatGPT models. We evaluate `Kimi K2.5` using `OpenHands` (Wang et al., 2025). We also evaluate `Claude Haiku 4.5`, `Gemini 3 Flash`, and `GPT-5.4 Mini` using `OpenHands` to compare performance across harnesses.

## 4.1. Multimodal Feedback

We describe two tooling configurations that allow agents to access richer multimodal information from Godot through editor screenshots and/or rendered video.

**Baseline.** As a baseline, each agent starts inside the project directory and is given the task instruction along with basic instructions on how to run Godot. We provide additional methods to support the agent, primarily to observe if additional visual context improves performance. We provide our full prompts in Appendix D.

**Editor Screenshot MCP.** We develop an MCP server that

loads the Godot editor for the current task, takes a screenshot of the editor, then returns the image to the agent. This allows the agent to view the game scene, the node tree, the node inspector, as well as other information present in the editor. This method allows the agent to leverage additional visual feedback to validate its solution.

**Runtime Video.** We provide agents with instructions on how to generate gameplay videos using Godot's built-in recording functionality, which is otherwise frequently ignored or misused. This differs from the MCP server as it captures both a) temporal elements only present in video and b) the current camera view (the editor does not show the camera view). Typically, models process videos into image frames using python rather than ingesting the video directly.

## 4.2. Discussion of Results

We now discuss our findings from evaluating agents on `GameDevBench` (Table 2).

**Game development proves challenging to even the most capable models and performance rapidly degrades when moving further from the frontier.** `GPT-5.4`, `Claude Sonnet 4.5`, and `Gemini 3 Pro` achieve baseline performances of 41.1%, 28.8%, and 50.1% respectively without additional multimodal feedback in their native agentic harness. Performance significantly degrades as we move further from the frontier. Without multimodal support, `Claude Haiku 4.5` solves 13.8%, `Kimi K2.5` solves 18.9%, and `Qwen3.5-397B` solves only 5.4% of tasks. In contrast, `Kimi K2.5` solves 91.3% of tasks in the frontend benchmark Design2Code (Si et al., 2024; BenchLM.ai, 2026), illustrating that strong open-weights multimodal performance on web UI tasks does not transfer to game development. Models struggle further on our `hard` subset of tasks which we discuss in more detail in Appendix H.

**Agent performance differs significantly across skill**

*Table 2.* Results from evaluating various models and agent harnesses on `GameDevBench`. Screenshot indicates that the agent was given access to an MCP server that screenshots the editor state. Video indicates that the agent was given additional instructions on how to generate a video of the current game scene. **Bold** and *italics* indicate the best and second best model performance.

| Harness | Model | w/ Screenshot | w/ Video | pass@1 (%) |
|---|---|:---:|:---:|---:|
| claude-code | claude-haiku-4-5-20251001 | ✗ | ✗ | 13.8 |
| | | ✓ | ✗ | 15.6 |
| | | ✗ | ✓ | 18.6 |
| | | ✓ | ✓ | 16.5 |
| | claude-sonnet-4-5-20250929 | ✗ | ✗ | 28.8 |
| | | ✓ | ✓ | 34.8 |
| codex | gpt-5.4-mini | ✗ | ✗ | 36.9 |
| | | ✓ | ✗ | 37.8 |
| | | ✗ | ✓ | 43.2 |
| | | ✓ | ✓ | 39.0 |
| | gpt-5.4 | ✗ | ✗ | 41.1 |
| | | ✓ | ✓ | *52.0* |
| gemini-cli | gemini-3-flash-preview | ✗ | ✗ | 45.4 |
| | | ✓ | ✗ | 45.4 |
| | | ✗ | ✓ | 46.9 |
| | | ✓ | ✓ | 44.1 |
| | gemini-3-pro-preview | ✗ | ✗ | 50.1 |
| | | ✓ | ✓ | **53.8** |
| openhands | claude-haiku-4-5-20251001 | ✗ | ✗ | 15.6 |
| | | ✓ | ✓ | 17.7 |
| | gpt-5.4-mini | ✗ | ✗ | 38.4 |
| | | ✓ | ✓ | 36.9 |
| | gemini-3-flash-preview | ✗ | ✗ | 30.3 |
| | | ✓ | ✓ | 31.8 |
| | kimi-k2.5 | ✗ | ✗ | 18.9 |
| | | ✓ | ✓ | 20.7 |
| | qwen3.5-397b-a17b | ✗ | ✗ | 5.4 |
| | | ✓ | ✓ | 5.1 |

**and editor categories.** We observe a general trend where agents perform worse on tasks that are more multimodally demanding (Figure 5). For skills, agents perform best at gameplay logic tasks (51.4%) and worst at 2D graphics and animation (33.0%) and UI (32.0%) tasks, which require agents to understand images or other assets for animations and effects. 3D graphics (38.4%) sit in between (scores are averages computed across the 8 agents in their native agentic harness with multimodal feedback enabled). Performance across editor categories is instead dependent on model capabilities. The models that achieve over 30% pass@1—GPT-5.4 Mini, GPT-5.4, Gemini 3 Pro, Gemini 3 Flash, and Claude Sonnet 4.5—perform similarly on tasks regardless of the required editor type. However, Claude Haiku 4.5, Kimi K2.5, and Qwen3.5-397B perform worse on tasks requiring the scene and contextual editors which are typically more multimodally demanding compared to scripting tasks.

**Multimodal tooling consistently improves agent performance.** We find that providing an agent with both the MCP and video instructions almost always improves performance. This trend holds across most models, though the gains vary across agents. We see the largest gain with GPT-5.4 where performance increases from 41.1% → 52.0%. The only exceptions to this are GPT-5.4 Mini in OpenHands—which is not its native harness—and Gemini 3 Flash where performance marginally decreases. Additionally, we evaluate GPT-5.4 Mini, Claude Haiku 4.5, and Gemini 3 Flash using the MCP or video instructions in isolation. Surprisingly, video runtime often provides more improvement than enabling both forms of multimodal feedback. An exception is Gemini 3 Flash which performs similarly across all methods. Although all tasks can be verified through code, visual feedback allows agents to verify and amend mistakes. This behavior strongly resembles that seen in recent work (Yin et al., 2026), where visual feedback improves agentic performance.

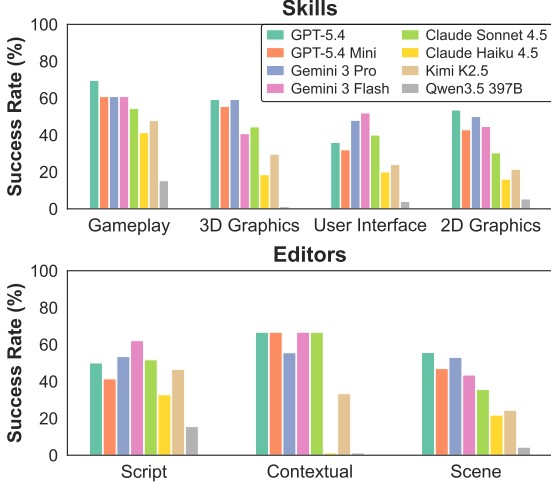

*Figure 5.* In general, agents perform better on tasks that require skills focusing on gameplay functionality compared to tasks that require multimodal understanding such as 2D and 3D graphics tasks. Performance across editor categories is dependent on the model: stronger frontier agents (`GPT-5.4 Mini`, `GPT-5.4`, `Gemini 3 Pro/Flash`, `Claude Sonnet 4.5`) maintain similar success across all editors, while weaker agents (`Claude Haiku 4.5`, `Kimi K2.5`, `Qwen3.5-397B`) drop sharply on the more multimodally demanding contextual and scene editors. All success rates are taken from results where the agent has access to multimodal feedback (MCP and Video).

**Agentic harness choice can impact performance, but the effect varies depending on the model.** We evaluate `Claude Haiku 4.5`, `Gemini 3 Flash`, and `GPT-5.4 Mini` using both their original harness and `OpenHands` (Table 2). When using `OpenHands`, we observe that `Claude Haiku 4.5` shows a small increase in performance from 16.5% to 17.7%. `GPT-5.4 Mini` offers mixed performance, improving over the baseline from 36.9% to 38.4%, yet degrading from 39.0% to 36.9% when using multimodal feedback in `OpenHands`. On the other hand, `Gemini 3 Flash`'s performance significantly decreases from 45.4% to 30.3% in `OpenHands`. This is likely due to incompatible editing tools between Gemini models and `OpenHands` [2].

**Cost varies significantly depending on the model, harness, and whether multimodal feedback is provided.** We find that enabling multimodal feedback almost always increases cost in exchange for increased performance (Figure 6). When enabling multimodal feedback, both `Gemini` and `Claude` models tend to show a relatively minor increase in costs. On the other hand, `GPT` models increase significantly in cost, with up to a 3.3x increase in `GPT-5.4` when enabling feedback. Interestingly, costs when using multimodal feedback does not increase significantly when `GPT-5.4 Mini` is used in `OpenHands`, although neither

---

[2]https://github.com/OpenHands/OpenHands/issues/9454

does performance. We find that `Gemini 3 Flash` in its native harness is the most cost-efficient model.

### 4.3. Error Analysis and Directions for Improvement.

We manually analyzed some of the most common errors that agents made when solving the task. These errors indicate potential gaps in capabilities and future directions for agent development. While there are a variety of errors, we observe two consistent error patterns. We provide a more detailed analysis on error patterns in Appendix I.

**Agents struggle with multimodal understanding.** Perhaps the most consistent error pattern occurs from a lack of multimodal understanding. Specifically, it is often necessary to understand multimodal inputs to properly complete a game development task. For example, creating (or even simply picking) an animation requires that the agent either parse through multiple images or pick out specific sprites within a spritesheet. Currently, agents frequently pick the wrong images or sprites (e.g., picking walking motion sprites instead of attacking motions). It is clear that improvements to multimodal understanding would significantly improve performance of agentic game development.

**Agents struggle with common game development patterns.** In game development, there are many common development patterns. For example, game elements (called nodes in Godot) form a tree structure where specific nodes such as an AnimatedSprite2D and CapsuleCollider handle animations and physics respectively. Another example would be signals that trigger between various files when conditions are met such as when two colliders intersect with each other. Agents frequently add nodes to incorrect levels in the tree, drop necessary signals, or assign resources to the wrong elements. We provide an example of such an error in Appendix G. This reinforces a long-standing trend within model training—models must be trained on specific domains to excel within that domain.

## 5. Related Works

**Agentic Benchmarks.** Software development has been one of the premier frontiers of agentic development. SWE-Bench (Jimenez et al., 2024; Yang et al., 2024a) was perhaps the first benchmark and catalyst towards agentic software development. Over time, multiple new software benchmarks have been developed (Chan et al., 2025; Merrill et al., 2026; Yang et al., 2025), but they remain largely unimodal. The few multimodal software benchmarks have largely focused on frontend JavaScript development (Zhu et al., 2025; Si et al., 2024; Yang et al., 2024b). Instead, the most common use case for multimodal agents has been computer use (Xie et al., 2024) and web navigation (Zhou et al., 2024; Koh et al., 2024). Progress in this domain is challenging as

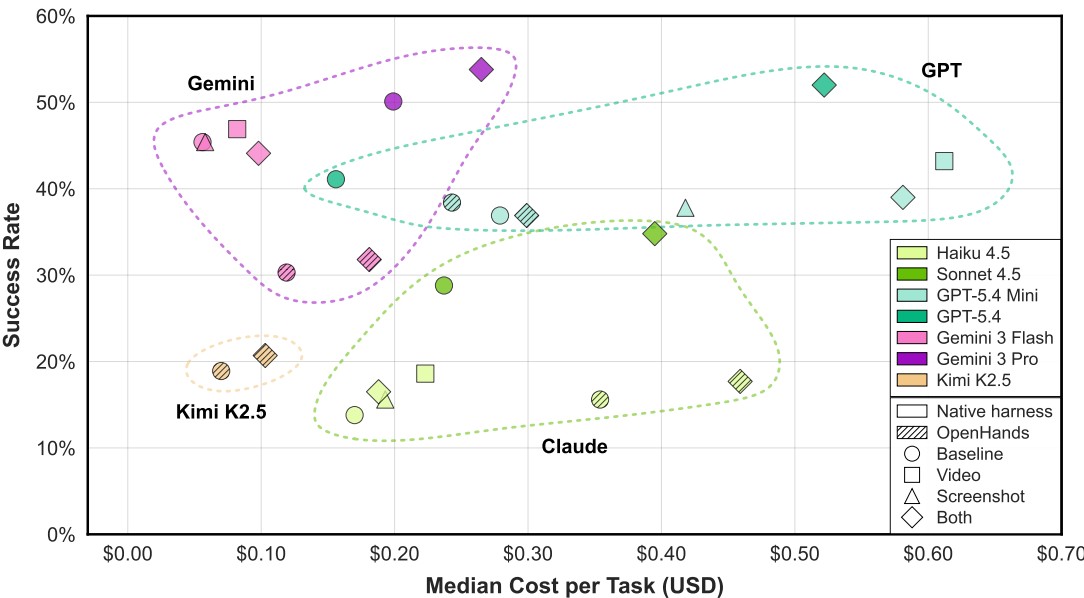

*Figure 6.* We capture the trade-off between performance and cost. We observe that Gemini models are the most cost efficient, offering good performance at lower costs. Among all the models, GPT costs vary the most when utilizing multimodal feedback. Comparatively, Claude models generally under-perform in both cost and performance.

agents must operate in an action space rather than simply writing code. Game development bridges the gaps between these domains by requiring multimodal input, but allowing for code output. `GameDevBench` is able to effectively reap benefits from both software and computer use domains, thus enabling effective multimodal evaluation.

**Game Playing.** There has always been significant interest in the application of artificial intelligence (AI) to games (Gallotta et al., 2024); gameplay has been seen as a proxy for the capabilities or intelligence of an AI system, with projects ranging from Deep Blue (Campbell et al., 2002), Alpha Go (Silver et al., 2016), and Cicero (FAIR et al., 2022) to more recent generalists such as SIMA 2 (Bolton et al., 2025). Practically, games provide an interactive simulation environment with clear reward signals allowing researchers to experiment with methods—particularly from reinforcement learning—to improve model capabilities. The recent flux of LLMs playing Pokémon (Karten et al., 2025a;b; Comanici et al., 2025) uses game-playing agents to evaluate and explore the agentic reasoning capabilities of frontier models which is then used directly in game development to test games (Nunu AI, 2024). This transition from game playing agents as NPCs and opponents in games to becoming a portion of the game development process marks a timely need for benchmarks such as `GameDevBench`.

**Game Development.** Concordia (Vezhnevets et al., 2023) and other subsequent work on tabletop role-playing games (Vezhnevets et al., 2025) seek to replace interactable characters with a highly adaptive story created entirely from interactions with LLMs. Other works try to fully replace the physics engine of the game to immediately generate frames based on player actions (Bruce et al., 2024). Procedural Content Generation has a long history of using AI for game asset creation (Summerville et al., 2018; Shaker et al., 2016) and evolutionary level design (Sudhakaran et al., 2023). However, these largely focus on a singular aspect of game development. Ultimately, each of these features still needs to be combined in a game engine to develop a full game, which is the capability `GameDevBench` directly evaluates.

## 6. Conclusion

We present `GameDevBench`, the first benchmark to evaluate an agent's ability to solve game development tasks. To create our benchmark, we develop a pipeline for converting YouTube and web tutorials into benchmark tasks. We find that agents struggle with tasks in game development, especially when tasks require deeper multimodal understanding. The gap between frontier and non-frontier models is sharp, with absolute differences of up to $48.7\%$ `pass@1`. Lastly, we show that even simple multimodal feedback tooling improves agent performance: when given access to screenshots and video, `GPT-5.4` increases from $41.1\%$ to $52.0\%$ `pass@1`, a $10.9$ percentage-point gain and a $26.5\%$ relative improvement. Our findings highlight the need to improve multimodal capabilities of agents—either through training or methods of visual feedback. We speculate addressing these needs would improve agentic performance in domains even beyond software and game development.

## Acknowledgements

This work was supported in part by the National Science Foundation grants IIS1705121, IIS1838017, IIS2046613, IIS2112471, the Department of Defense (DoD) through the National Defense Science and Engineering Graduate (NDSEG) Fellowship Program, and funding from Datadog. Any opinions, findings and conclusions or recommendations expressed in this material are those of the author(s) and do not necessarily reflect the views of any of these funding agencies.

## Impact Statement

This work introduces `GameDevBench`, a benchmark for evaluating agentic capabilities in game development, a domain that combines large-scale software engineering with rich multimodal reasoning.

**Positive impacts.** `GameDevBench` may accelerate the development of more capable and robust multimodal agents, with downstream benefits extending beyond game development to other visually grounded, context-dense domains such as robotics, simulation, design tools, and interactive software engineering. Additionally, by focusing on an open-source engine (Godot) and releasing tasks and infrastructure publicly, this work lowers barriers to entry and supports reproducible research by both academic and independent researchers.

**Potential risks and misuse.** Game development is both a software discipline and a creative discipline. Improved agentic capabilities could contribute to workforce disruption in creative and technical fields either directly in game development or beyond. While `GameDevBench` does not aim to replace human developers, advances driven by this benchmark may enable partial automation of tasks traditionally performed by artists or programmers.

**Mitigations and ethical considerations.** To reduce potential risks, tasks are derived from permissively licensed tutorials, and no proprietary assets or data are included. Additionally, none of the tasks involve *creating* assets themselves, instead focusing exclusively on tasks that involve the use of *pre-existing* assets, thus limiting direct impact to artists and creatives.

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

# A. Task Construction Prompt

Below is the full prompt provided to the Codex agent for automatic task construction from YouTube tutorials (Stage 2). The agent receives this prompt along with a pointer to a specific tutorial folder containing a video transcript, metadata, and a GitHub repository URL.

---

**Task Construction System Prompt**

```
# YouTube Tutorial to Task Construction Guide

This guide explains how to convert a single YouTube Godot tutorial (with transcript and GitHub repo) into
GameDevBench tasks.

GameDevBench is a multimodal LLM Agent benchmark to test if models can develop games or assist with game
development.

## Godot

Godot is installed and usable with the `godot` command.

VERY IMPORTANT: Whenever you run `godot` please ensure you set a timeout of 1 minute.

## Context

You will be working in a single tutorial folder at a time. Each folder contains:

  {data_folder}/{channel_name}/{video_title}/
  +-- transcript.txt      # Full video transcript
  +-- metadata.json       # Video metadata
  +-- github_repo.txt     # GitHub repository URL

### Tips for YouTube Tutorial Processing

1. Transcript Context: Tutorials often explain "why" before "what" - look for action verbs
2. GitHub is Ground Truth: When transcript is unclear, GitHub repo shows what actually works
3. Simplify Complexity: If tutorial covers multiple concepts, break into multiple tasks
4. Test Repository First: Clone and run GitHub repo to understand expected behavior
5. Match Repo Structure: Use similar node names and organization as the repo
6. License Compliance: All repos already filtered for MIT/Apache-2.0/CC0-1.0

### Common Pitfalls

- Copying GitHub Repo Verbatim: Adapt to GameDevBench structure, don't just copy
- Ignoring Transcript: GitHub shows "what" but transcript explains "why" and learning objective
- Overly Broad Tasks: Focus on one specific learning objective per task
- Missing Assets: Ensure sprites/sounds from repo are included in both task directories
- Weak Validation: Check everything that makes the task correct

### Key Principles for Single-Folder Processing

1. All analysis happens in the tutorial folder first
    - Clone repo to repo/ subdirectory
    - Create analysis_progress.md for documentation
    - Complete all analysis before creating tasks
2. Document everything as you go
    - Update analysis_progress.md after each step
    - Include transcript quotes, repo structure, task ideas
    - Track what works and what doesn't
3. Test the GitHub repo before extracting tasks
    - Run godot --import-all --quit
    - Verify it's a working Godot project
    - Check for missing assets or dependencies
4. Navigate to GameDevBench root for task creation
    - Don't create tasks inside the tutorial folder
    - Copy assets from tutorial's repo/ to task directories
5. Return to tutorial folder for final documentation
    - Update analysis_progress.md with completion status
    - Note which tasks were created
    - Record any issues for future reference

## Phase 1: Setup

### Step 1: Check for Godot 4.

We only want to operate on Godot 4 tutorials. If the tutorial folder / github repo is for a Godot 3 project,
stop and report that.
```

```
### Step 2: Set Up Your Workspace

Create a progress tracking file to document your work.

### Step 3: Read Available Files

Read the transcript, GitHub repo URL, and metadata. Document in analysis_progress.md:
- Main topic, key concepts, estimated complexity
- Has implementation steps: yes/no
- GitHub repo available: yes/no
- Suitable for tasks: yes/no/maybe

### Step 4: Analyze Transcript for Task Ideas

Look for in transcript.txt:
- Node creation mentions ("create a CharacterBody2D")
- Node adjustments ("adjust the anchors of the container")
- Property settings ("set the gravity to 980")
- Script attachment steps ("attach a new script")
- Signal connections ("connect the body_entered signal")
- Scene organization instructions
- Multimodal reasoning ("create an animation from the spritesheet")

Task Categories:
- Graphics & Animation
- Physics & Movement
- World Building
- Programming
- User Interface
- Game Systems
- Audio

IMPORTANT:
- The skillset required between tasks should be diverse. Focus on tasks that require adjusting node properties,
adding new nodes, or adding sub-children.
- Focus on keeping the tasks faithful to the tutorial.
- Each task must be independent from each other.
- Tasks that require multimodal reasoning (e.g., cutting up a spritesheet, adjusting sound to match animation)
are especially desirable.

### Step 5: Clone and Examine GitHub Repository

Clone to a repo/ subdirectory. Examine structure, key files, project.godot for Godot version. Document:
- Main scene, key scenes, scripts, assets
- Key nodes and configuration
- Critical properties
- Scripts summary

Create a dependency graph mapping file and feature dependencies.

### Step 6: Test the GitHub Repository

Import assets, try to run the project, check for tests. Document import results, project status, and conclusion.

## Phase 2: Task Creation

### Step 1: Extract Actionable Tasks

Based on transcript + GitHub repo, identify specific, testable tasks. Ensure tasks center on node creation and/
or inspector configuration.

Difficulty Guidelines:
- Easy: 1 to 3 individual steps
- Medium: 4 to 8 individual steps
- Hard: 9 or more individual steps

For each task document:
- Source (transcript line + repo file)
- What to create (specific nodes/properties)
- Validation criteria
- File modifications from start to finish state
- Difficulty, GitHub reference, categories, multimodal flag

### Step 2: Create Task Directories in GameDevBench

Navigate to GameDevBench root. Determine next task number. Create directories for both tasks/ and tasks_gt/.
Copy project template and needed assets from cloned repo.

### Step 3: Create task_config.json
```

```
{
  "task_id": XXXX,
  "name": "Descriptive Task Name from Video",
  "instruction": "Clear, specific instructions...",
  "difficulty": "easy|medium|hard",
  "template_id": X,
  "metadata": {
    "tutorial_folder": "...",
    "tutorial_source": "YouTube: {channel} - {video_title}",
    "video_id": "...",
    "github_repo": "...",
    "transcript_excerpt": "...",
    "expected_nodes": ["NodeType1", "NodeType2"],
    "key_properties": {"property": "expected_value"}
  },
  "tags": ["youtube", "2d|3d", "category", "node-type"]
}
```

### Step 4: Reference GitHub Repo for Ground Truth

- GitHub repo shows the completed task
- Adapt to fit GameDevBench structure (don't copy verbatim)
- Document which files were copied and which were not

### Step 5: Create Ground Truth Implementation

Study GitHub repo scene structure, recreate key nodes and hierarchy, set properties, add scripts, simplify if needed, and test.

## Phase 3: Task Instruction

### Step 1: Create Task Instruction

Key principles:
- Concise, clear, and unambiguous
- Solver must understand requirements to go from start state to ground truth
- Solver will NOT have access to tests or ground truth
- Self-contained: no references to transcript, tests, other tasks, or the tutorial name
- Mention technical requirements (node types, APIs) but not usage details
- NO tips, NO hints, NO test commands, NO code examples

Each instruction step must have evidence pointing back to the original source (transcript or repository).

## Phase 4: Task Validation

### Step 1: Create Validation Script

Create a GDScript validation that:
- Asserts required nodes exist in the correct hierarchy
- Confirms critical inspector values left unset in the starting point
- Fails early when structural requirements are missing
- Prints VALIDATION_PASSED or VALIDATION_FAILED messages
- Copy to BOTH task and ground truth directories

Document how each test maps to a specific instruction step.

### Step 2: Create Starting Point (Incomplete Version)

- Provide basic scaffolding so the scene launches
- Include required raw assets
- Omit key implementation details:
  - Leave tutorial-created nodes absent
  - Skip scripts and signal connections
  - Leave tutorial-modified inspector properties unset

Goal: Starting point should fail validation but provide foundation.

### Step 3: Test and Validate

- Starting point: should output VALIDATION_FAILED
- Ground truth: should output VALIDATION_PASSED

## Final Quality Checklist

- analysis_progress.md fully completed
- Each instruction step lists transcript or repo evidence
- Validation maps to instruction with file + line numbers
- Multiple independent tasks created from the tutorial

```
- Every task has both ground truth and starting point
- Starting points fail validation
- Ground truths pass validation
- Each task includes valid main.tscn and test.tscn
- Node/inspector-focused requirements asserted
- At least one task requires multimodal reasoning
- Asset transfer summary recorded
- Deviations, blockers, and Godot version issues documented
```

## B. Task Refinement Prompt

Below is the full prompt provided to the agent for automatic task validation and refinement (Stage 3). The prompt consists of two parts: (1) an instruction that describes the validation workflow and context, and (2) a checklist template that the agent must fill out with evidence for each criterion. If any criterion fails, the agent is instructed to fix the task accordingly.

A variant of this prompt omits scripting-related checks and adds the constraint "No .gd script editing is required," which was used for tasks that focus exclusively on scene construction and inspector configuration.

**Task Refinement System Prompt**

```
# Benchmark Task Validation Guide

This file documents instructions to validate a task for GameDevBench.

GameDevBench is a multimodal LLM Agent benchmark to test if models can develop games or assist with game
development.

## Context

You will be working on a single task at a time. Each task has three related folders:

### Tutorial Folder
  {data_folder}/{channel_name}/{video_title}/
  +-- repo/                  # YouTube tutorial repository
  +-- analysis_progress.md   # Notes from task generation
  +-- transcript.txt         # Full video transcript
  +-- metadata.json          # Video metadata
  +-- github_repo.txt        # GitHub repository URL

### Task (Starting Point) Folder
  tasks/task_{number}_{name}/
  +-- task_config.json       # Contains the task instruction
  +-- scripts/test.gd        # Contains the test code

### Task (Ground Truth) Folder
  tasks_gt/task_{number}_{name}/
  (Same structure as starting point, with completed solution)

## Instructions

Your job is to:
1. Read and analyze the transcript
2. Read and examine the GitHub repository
3. Read and examine the task created
4. Document your progress
5. Validate whether the task satisfies each criterion
5a. Each criterion must have evidence for validation documented

Copy the checklist template into the task starting point folder and fill it out as you validate.

---

# Key Checklist

- [ ] The task starting point runs with `uv run gamedevbench validate $TASK_NAME` and successfully outputs a
test failure.
  - Evidence:

- [ ] The task ground truth runs with `uv run gamedevbench --gt validate $TASK_NAME` and successfully outputs
SUCCESS.
```

```
   - Evidence:

- [ ] In every task, there exists a valid main.tscn and test.tscn similar to tasks_gt/task_0001_place_asset.
   - Evidence:

- [ ] The task instruction matches the tutorial transcript. The task instructions must be a subset of the
tutorial transcript.
   - Instruction 1 / Transcript 1
   ...
   - Instruction N / Transcript N
   - Instructions Missing from Transcript: Explanation

- [ ] The task code is directly derived from the repository code. Please document where the derived code is.
   - Evidence:

- [ ] The task instruction is clear, unambiguous, and self-contained. There are no references to the tutorial or
 other tasks.
   - Evidence:

- [ ] The tests in test.gd match the instructions. All tests are contained in the instruction. Similarly, all
instructions are in the tests. Explain how to adjust the tests themselves to match the instructions.
   - Test 1, Instruction 1
   - Test 2, Instruction 2
   ...
   - Test N, Instruction N
   - Missing Coverage: Explanation here

- [ ] Each test in test.gd is unambiguously defined in the instructions. With just the instruction and the task
code (without looking at the tests), it is unambiguously possible to satisfy each test condition.
   - For EACH test, list EVERY assertion/check it makes, then verify the instruction specifies that EXACT detail:
   - Test 1, Assertions: [list each check], Instruction coverage: [exact instruction text]
   - Test 2, Assertions: [list each check], Instruction coverage: [exact instruction text]
   ...
   - CRITICAL AMBIGUITY CHECKS - For each test, verify:
     - [ ] String formatting (padding, delimiters, exact format) is specified in instruction
     - [ ] Exact string values/names are in instruction
     - [ ] Number formats (zero-padding, decimal places) are specified
     - [ ] Any comparison operators have clear criteria
     - [ ] Node names, paths, and types match instruction
     - [ ] Property values (numbers, booleans, strings) have exact values in instruction
   - Ambiguous Tests: [List any test checks that lack exact specification in instruction]

- [ ] If there are multiple solutions to the problem, the tests in test.gd are flexible to allow multiple
solutions. Mark as completed if there is only one solution and that solution is clearly decipherable from the
instructions.
   - Evidence:

- [ ] The folder and file names are consistent with other tasks (tasks_gt/task_0001_place_asset).
   - Evidence:

- [ ] PROCEED. Check this box if the task is validated and all key checks pass successfully.

# Feature Checklist

- [ ] The task contains instructions or goals that are Node/inspector-focused.
   - Evidence:

- [ ] The task contains or requires multimodal reasoning or understanding to complete.
   - Evidence:

- [ ] The task contains a multimodal input (such as an image) in the instruction.
   - Evidence:

# Identifying Ambiguous Tests (Examples)

## Example 1: AMBIGUOUS - Vague formatting requirement

Instruction: "Format incremental and timer step text"

Test Code:
  var expected_text = "Destroy 5 ships 00/05"
  if do_label.text != expected_text:
```

```
        issues.append("Initial text should be '%s'" % expected)

Assertions:
- Checks text equals exactly "Destroy 5 ships 00/05"
- Requires zero-padded format
- Requires specific spacing and delimiter

Why AMBIGUOUS: Instruction says "format text" but does not specify zero-padding, exact delimiter, spacing, or
format string structure.

How to fix: Change instruction to "Format incremental step text as '{details} {collected:02d}/{required:02d}'"
OR make test flexible to accept any reasonable format.

## Example 2: UNAMBIGUOUS - Specific requirement

Instruction: "Set the ColorRect size to Vector2(screen_max_size, screen_max_size)"

Test Code:
  var expected = Vector2(screen_max_size, screen_max_size)
  assert(color_rect.size == expected)

Why UNAMBIGUOUS: Instruction explicitly states the exact Vector2 formula. No ambiguity about what value is
expected.

## Example 3: AMBIGUOUS - Missing specific values

Instruction: "Connect to QuestManager signals"

Test Code:
  var required = ["step_updated", "step_complete", "quest_completed", "quest_failed"]
  for signal_name in required:
      if not _has_connection(qm, signal_name, quest_ui):
          issues.append("Must connect to %s" % signal_name)

Why AMBIGUOUS: Instruction says "signals" (vague) but test checks for 4 specific signal names. A solver might
connect only 2 signals and technically satisfy "connect to signals".

How to fix: Change instruction to "Connect to QuestManager signals: step_updated, step_complete, quest_completed
, and quest_failed".
```

## C. Human Annotation Instructions

Below are the instructions provided to human annotators during Stage 4. Annotators were asked to verify task correctness, fix common issues, and flag tasks that were unsalvageable.

**Human Annotation Instructions**

```
# Human Validation

Goal: Ensure that tasks are solvable and not ambiguous. Essentially, we want to ensure that tasks are actually
good tasks.

Fixing a task should take 5-15 minutes per task at most. If it takes longer, then it may be too difficult for
us to fix, in which case we can skip the task and mark it as Blocked in the Status.

## Common Issues

Most tasks require minor fixes. By far the most common are the following:

- Ambiguous Instructions (e.g., model says to change the skybox, but there may be three different skybox
variables possible)
- Overly Strict Tests (e.g., tests require something that isn't stated in the instruction)
  - Naming is a good one (tests for a named node, without requesting it)
- Conflicting Instructions (e.g., model says to do two incompatible things)
- Tutorial References - sometimes the instruction mentions the tutorial; just remove that reference

As with previous, the best way to find these errors is to actually implement the task. However, in this case
you should feel free to use any tool (e.g., a coding agent directly) to support you or implement the task. The
goal is NOT to see if you can solve the task (as it was before), but to identify errors in the task.

## General Procedure
```

1. Look at the task (base and ground truth versions) in the editor. Read the task instruction. See if it looks reasonable (multimodality) or if it's clearly a scripting oriented task. What you're looking for is something that just makes sense. Run this before to ensure everything loads properly:
    godot --path /path/to/folder --editor
2. Change directory to the task. Ask your agent of choice to solve the task.
3. While the agent is solving the task, take a look at test.gd. See if each test matches the instruction. If not, fix. You can usually catch some easy errors, such as named node/function tests.
4. After the agent finishes, run validation. See if it passes / fails.
5. If the agent failed, pass the test.gd in. Ask the agent if it missed anything. Ask the agent if the things it missed are due to its own error or due to ambiguous instruction/overly strict tests. This is my usual prompt :
    "Look at test.gd. Was there anything you missed? Did you miss it due to your mistake or were the instructions unclear?"
   The agent is sometimes good, sometimes bad at this. It really depends on the agent.

## Multimodality

You may encounter tasks that have poor multimodality. If it's easy to fix, then just fix it (e.g., something is not in the camera, some default is incorrect). If it's not, flag it and move on. Note that some tasks are inherently multimodal in which case don't worry about it. We have an automated script that is improving multimodality, so really just fix any obvious errors.

## Skipping Tasks

If the project doesn't play, doesn't load, or some other catastrophic issue, just skip the task. There should be very few tasks like this.

Please mark all the issues using the issues dropdown. This is important so we can double back to tasks if necessary.

## Important: Preventing Agent Contamination

When using an agent to attempt the task, always run the task in the task folder itself and instruct the model:
- Don't look at test.gd, test.tscn or execute any tests
- Don't look outside of your current folder
- Don't look at task_config.json
- Don't look at task_validation.md

## D. Prompt Templates

All task prompts are derived from the same base instruction, with optional extensions that provide additional multimodal feedback mechanisms. We report the exact prompt templates used for the baseline, MCP-enabled, and runtime-video-enabled settings below.

---

**Baseline Prompt**

```
<INSTRUCTION FROM task_config.json>

You must complete the full task without any further assistance.
Godot is installed and you can run godot using the `godot` command.
It is recommended to run this with a timeout (e.g., `timeout 10 godot`)
to prevent hanging.
You are a visual agent and can use images and videos to help you
understand the state of the game.
```

---

**Prompt with MCP**

```
<INSTRUCTION FROM task_config.json>

You must complete the full task without any further assistance.
Godot is installed and you can run godot using the `godot` command.
It is recommended to run this with a timeout (e.g., `timeout 10 godot`)
to prevent hanging.
You are a visual agent and can use images and videos to help you
understand the state of the game.

You have access to a Godot MCP (Model Context Protocol) server that
provides specialized tools for working with Godot projects.

Available MCP Tools:
- `godot-screenshot`: Takes a screenshot of the Godot editor.

Usage Guidelines:
- Use screenshots before starting to understand project structure.
- Use screenshots after making changes to verify correctness.
- Use screenshots during debugging to inspect editor state.
```

---

**Prompt with Runtime Video**

```
<INSTRUCTION FROM task_config.json>

You must complete the full task without any further assistance.
Godot is installed and you can run godot using the `godot` command.
It is recommended to run this with a timeout (e.g., `timeout 10 godot`)
to prevent hanging.
You are a visual agent and can use images and videos to help you
understand the state of the game.

You can run the game and capture visual output using:
- `godot --path . --quit-after 1 --write-movie output.png`

You can capture short videos using:
- `timeout 60s godot --path . --quit-after 60 --write-movie output.avi`

Ensure that Godot exits after execution to avoid hanging.
Use images or videos to verify that your changes worked as expected.
```

# E. Task Examples

We provide examples of tasks in `GameDevBench`. Each task can be solved by taking actions in the editor as a human would or by directly editing code files.

## E.1. Isometric Crusader Animation

In this example, the goal is to add physical collision and animation to the character. This is a **2D graphics and animations** task that focuses on the animation editor which is a **contextual editor**.

*Figure 7.* An example task from `GameDevBench`. In this example, the goal is to add physical collision and animation to the character. This can be achieved through either taking actions directly in the editor or editing code files. Each action in the editor is equivalent to specific modifications within the code files. Matching steps are denoted with the same numbers in our figure.

## E.2. Floating Balls

In this example, the goal is to populate an empty 3D scene with a water depth visualization, including environment lighting, shader-driven water plane, background spheres, and a camera. This is a **3D graphics and animations** task that focuses on the **scene editor**.

> **Instruction:** Populate res://scenes/main.tscn so Main contains a water depth visualization scene: add a WorldEnvironment with a procedural sky and glow enabled, a DirectionalLight3D with shadows enabled, a MeshInstance3D named Water that uses a 10x10 PlaneMesh subdivided 20x20 with a ShaderMaterial pointing at res://scenes/WaterShader.tres, a Background Node3D holding exactly five sphere MeshInstance3Ds positioned near the water surface (y between -2 and 2) with green StandardMaterial3D overrides, and a Camera3D positioned at (-4.269, 0.54, -3.258) with a 70.9° FOV and all spheres are visible.

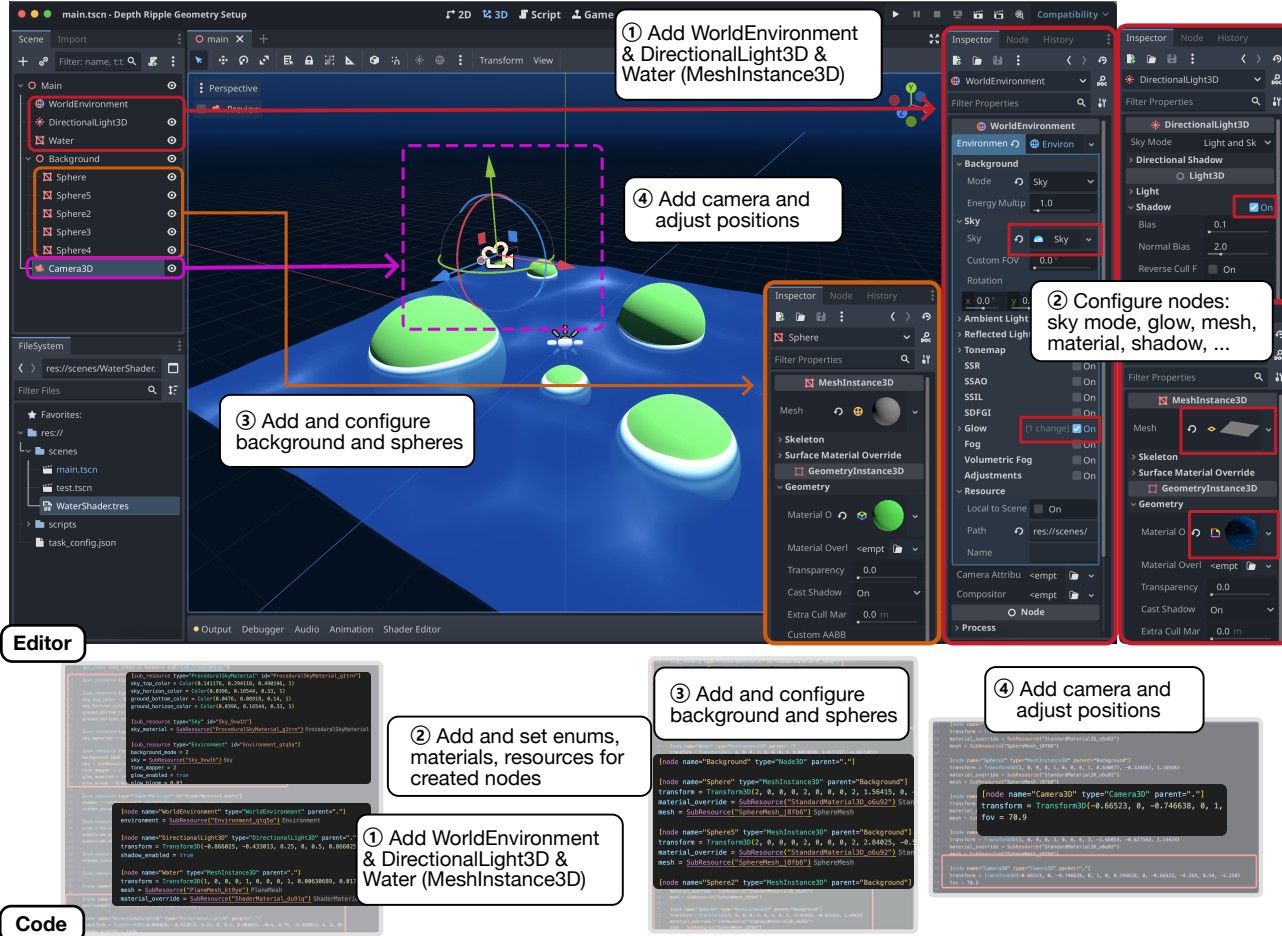

*Figure 8.* An example task from `GameDevBench`. In this example, the goal is to populate an empty 3D scene with a water depth visualization, including environment lighting, shader-driven water plane, background spheres, and a camera. This can be achieved through either taking actions directly in the editor or editing the scene file (`main.tscn`). Each action in the editor is equivalent to specific modifications within the scene file. Matching steps are denoted with the same numbers in our figure

## E.3. FPS User Interface

In this example, the goal is to build a complete three-screen menu system (Launch, Pause, and Restart) and signal connections to the menu handler script. This is a **user interface** task that focuses on the **scene editor**.

**Instruction**: Create res://scenes/menus.tscn as a Control named Menus that fills the viewport, keeps process mode Always, and uses scripts/menu_handler.gd. **Add three hidden panels, all centered with anchors at 0.5: LaunchMenu (offsets: -205, -196 to 205, 196), PauseMenu (offsets: -205, -196 to 205, 45), and RestartMenu (offsets: -205, -196 to 205, 63). Give LaunchMenu a VBoxContainer with 'FPS Horror' / 'By Bonkahe' labels; give PauseMenu a 'Paused' label; and give RestartMenu a 'You Died' label. Add the appropriate buttons to each (FullScreenButton, PlayButton, QuitButton, ResumeButton, RestartButton) using 31pt font overrides and exact node names. Finally, add a CanvasLayer on layer 2 with a TransitionOverlay ColorRect that ignores mouse input and stretches to the viewport**; assign assets/materials/menu_overlay_material.tres to it, export the LaunchMenu/PauseMenu/DeathMenu references on the root, and **connect every button's pressed signal to the proper menu handler method (ToggleFullScreen, BeginLaunch, HidePauseMenu, RestartLevel, ExitGame)."**

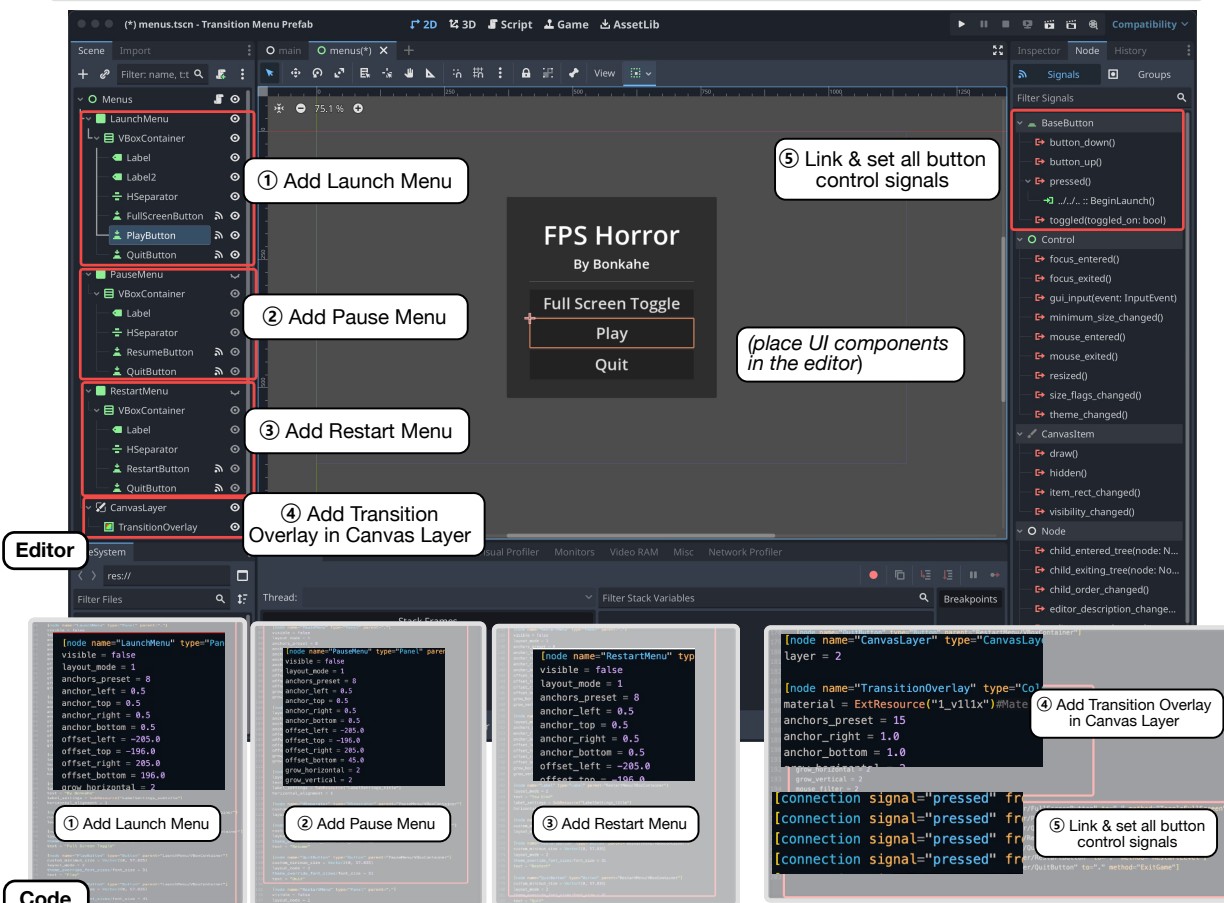

*Figure 9.* An example task from GameDevBench. In this example, the goal is to build a complete three-screen menu system (Launch, Pause, and Restart) with styled buttons, title labels, a shader-driven transition overlay, and signal connections to the menu handler script. This can be achieved through either taking actions directly in the editor or editing the scene file (menus.tscn). Each action in the editor is equivalent to specific modifications within the scene file. Matching steps are denoted with the same numbers in our figure.

## E.4. RTS Unit

In this example, the goal is to build a reusable RTS unit with a sprite, collision shapes, a detection area for neighbor avoidance, and an aura shader that highlights the unit when selected. The main focus is on the scripting, thus this is a **gameplay logic** task that focuses on the **script editor**.

**Instruction**: Build the reusable Unit scene: root it in a CharacterBody2D on collision layer 2/mask 3, keep it in the 'units' group. Add a Sprite2D child that uses the towerDefense spritesheet texture with region_enabled=true and region_rect=Rect2(960, 640, 64, 64), and apply a ShaderMaterial using the aura.gdshader. Add a CollisionShape2D child with a CircleShape2D of radius 14. Add an Area2D child named 'Detect' (collision layer 2, collision mask 2) with its own CollisionShape2D child using a CircleShape2D of radius 35. Implement unit.gd to export a speed variable, define a target_radius variable, include set_selected() and set_target() setters, an avoid() function that uses $Detect.get_overlapping_bodies(), and use move_and_collide() for movement. The set_selected() function must toggle the aura_width shader parameter (1.0 when selected=true, 0.0 when selected=false). Duplicate the Sprite2D material in _ready() to ensure each instance has its own material.

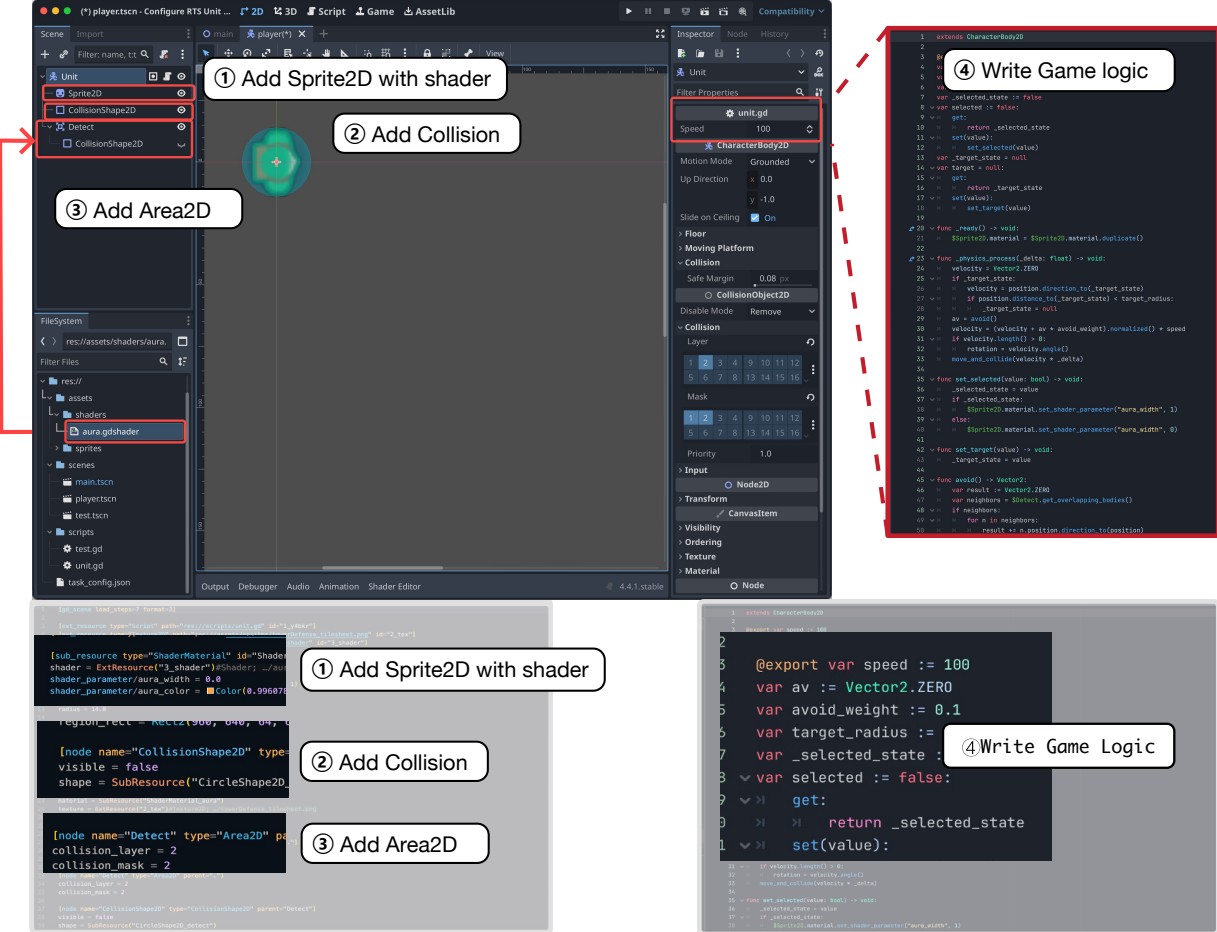

*Figure 10.* An example task from `GameDevBench`. In this example, the goal is to build a reusable RTS unit with a sprite, collision shapes, a detection area for neighbor avoidance, and an aura shader that highlights the unit when selected. Unlike purely scene-based tasks, this task requires both editing the scene file (`player.tscn`) and implementing gameplay logic in a script file (`unit.gd`). Each action in the editor is equivalent to specific modifications within the code files. Matching steps are denoted with the same numbers in our figure.

## F. Task Statistics

We provide detailed statistics for `GameDevBench`. Different tasks test different skills resulting in drastically different data distributions. For example, some sprite animation tasks require thousands of sprites to be processed.

*Table 3.* Comprehensive GameDevBench Task Statistics. Mean ($3\sigma$) denotes the mean after excluding values more than 3 standard deviations from the mean.

|  |  | **Mean** | **Mean** ($3\sigma$) | **Median** | **Max** |
|---|---|---|---|---|---|
| Overview | Files | 67.7 | 18.7 | 10.0 | 1929 |
|  | Filetypes | 6.7 | 6.6 | 6.0 | 18 |
|  | Lines of Code | 518.9 | 392.7 | 204.0 | 20072 |
|  | Nodes | 19.7 | 13.5 | 7.0 | 982 |
| Godot Scripting | Scripting Files | 3.0 | 2.4 | 2.0 | 49 |
|  | Scripting Lines | 236.7 | 176.6 | 107.5 | 9543 |
| Godot Scenes | Scene Files | 3.6 | 3.2 | 3.0 | 54 |
|  | Scene Lines | 219.5 | 154.6 | 32.5 | 10282 |
| Assets | Images | 53.4 | 4.3 | 1.0 | 1920 |
|  | Image Size (px) | 119.5K | 73.3K | 71.8K | 16.8M |
|  | Shaders | 0.2 | 0.1 | 0.0 | 7 |
|  | Audio | 1.0 | 0.2 | 0.0 | 13 |
|  | Resources | 0.7 | 0.4 | 0.0 | 14 |
| Gold Patch | Files Edited | 4.7 | 4.5 | 4.0 | 17 |
|  | Filetypes Edited | 3.2 | 3.2 | 3.0 | 6 |
|  | Total Lines Edited | 114.1 | 70.2 | 48.5 | 1949 |
|  | Scripting Lines Edited | 14.9 | 11.3 | 0.0 | 208 |
|  | Scene Lines Edited | 92.2 | 47.5 | 24.0 | 1949 |
|  | Nodes Edited | 2.3 | 1.9 | 1.0 | 24 |

# G. Case Study of Model Failure

## G.1. Common Game Development Patterns

Figure 11 shows a representative failure of common game development. The task requires completing a Godot `.tscn` scene file for a rain particle system, including wiring the `sub_emitter` property on a `GPUParticles2D` node to a sibling `Splash` node. `GPT-5.4` produces the correct property name and value (`sub_emitter = NodePath("../Splash")`), but places it under the `ParticleProcessMaterial` sub-resource instead of the `GPUParticles2D` node. The `sub_emitter` property belongs to `GPUParticles2D` and has no meaning on a material resource, indicating that the model lacks the knowledge that this property must be placed under the `GPUParticles2D` node.

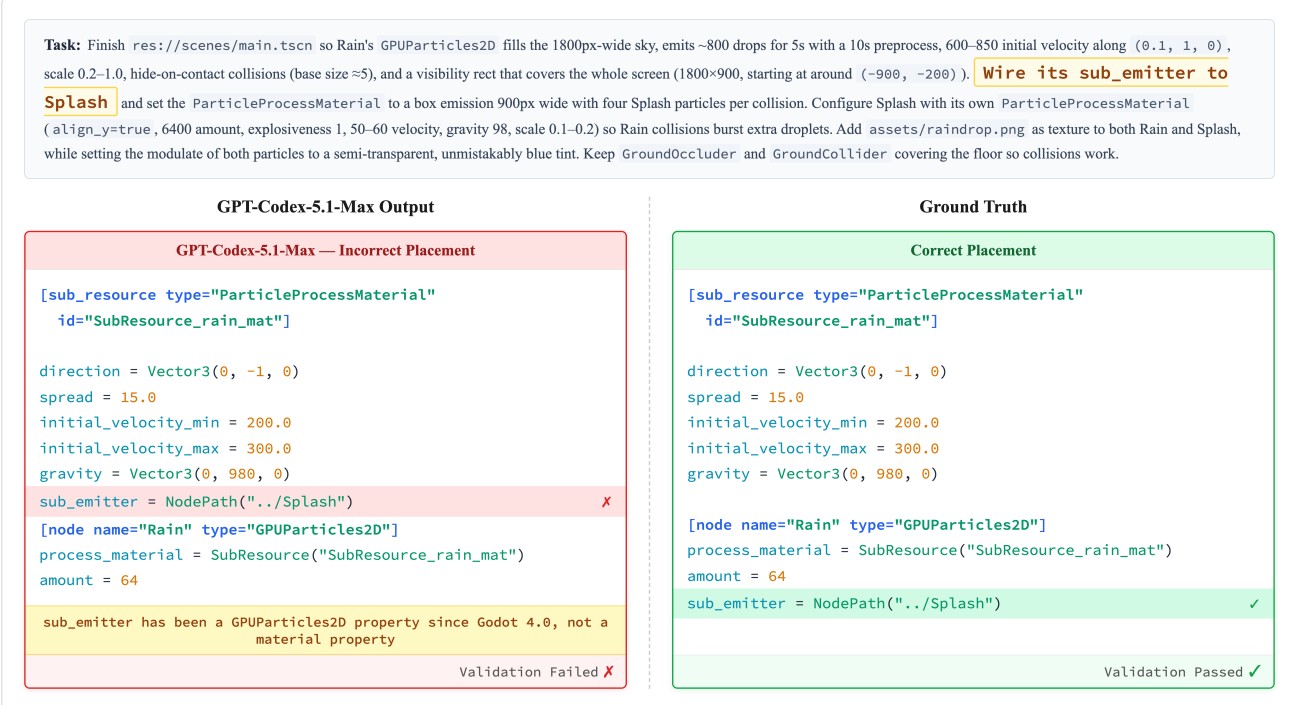

*Figure 11.* Example of Godot common game development task. `GPT-5.4` places `sub_emitter` inside the `ParticleProcessMaterial` sub-resource (left, red) instead of on the `GPUParticles2D` node (right, green). The property belongs to `GPUParticles2D`.

# H. Detailed Benchmark Results

## H.1. Full Benchmark Result with Token Count and Cost

Table 4 reports detailed benchmark results for both the core task set and the full task set(containing variant tasks deriving from core tasks). Overall, stronger models generally achieve higher pass@1, and multimodal inputs provide gains in performance across models. We also include average token usage and cost to provide a reference for comparing effectiveness and efficiency across agent and model setups.

*Table 4.* Core tasks indicates the tasks we build from the pipeline in Section 2. Full tasks also include task variations derived from core tasks. Tokens and costs are per-task medians over all records contributing to each row (k = 1,000 tokens). Task variants were designed to explicitly test for multimodal understanding. Full pass@1 is lower due to this increased requirement for multimodal understanding. Trends in general hold, except GPT-5.4 and Gemini 3 Pro swap first and second place.

| Harness | Model | w/ Screenshot | w/ Video | Core pass@1 (%) | Full pass@1 (%) | Median Tokens | Median Cost ($) |
|---|---|---|---|---|---|---|---|
| claude-code | claude-haiku-4-5-20251001 | ✗ | ✗ | 20.8 | 13.8 | 10.5k | 0.170 |
| | | ✓ | ✗ | 21.4 | 15.6 | 5.7k | 0.193 |
| | | ✗ | ✓ | 25.3 | 18.6 | 11.6k | 0.223 |
| | | ✓ | ✓ | 24.7 | 16.5 | 10.3k | 0.188 |
| | claude-sonnet-4-5-20250929 | ✗ | ✗ | 31.2 | 28.8 | 6.6k | 0.237 |
| | | ✓ | ✓ | 41.6 | 34.8 | 8.2k | 0.395 |
| codex | gpt-5.4-mini | ✗ | ✗ | 48.1 | 36.9 | 329k | 0.279 |
| | | ✓ | ✗ | 46.1 | 37.8 | 546k | 0.418 |
| | | ✗ | ✓ | 52.6 | 43.2 | 848k | 0.612 |
| | | ✓ | ✓ | 48.7 | 39.0 | 786k | 0.581 |
| | gpt-5.4 | ✗ | ✗ | 49.4 | 41.1 | 188k | 0.156 |
| | | ✓ | ✓ | **56.5** | *52.0* | 865k | 0.522 |
| gemini-cli | gemini-3-flash-preview | ✗ | ✗ | 47.4 | 45.4 | 296k | 0.056 |
| | | ✓ | ✗ | 48.1 | 45.4 | 284k | 0.058 |
| | | ✗ | ✓ | 51.9 | 46.9 | 464k | 0.082 |
| | | ✓ | ✓ | 50.0 | 44.1 | 582k | 0.098 |
| | gemini-3-pro-preview | ✗ | ✗ | 48.7 | 50.1 | 329k | 0.199 |
| | | ✓ | ✓ | *54.5* | **53.8** | 492k | 0.265 |
| openhands | claude-haiku-4-5-20251001 | ✗ | ✗ | 22.7 | 15.6 | 1,433k | 0.354 |
| | | ✓ | ✓ | 27.9 | 17.7 | 1,853k | 0.459 |
| | gemini-3-flash-preview | ✗ | ✗ | 31.8 | 30.3 | 350k | 0.119 |
| | | ✓ | ✓ | 36.4 | 31.8 | 583k | 0.181 |
| | gpt-5.4-mini | ✗ | ✗ | 44.8 | 38.4 | 809k | 0.243 |
| | | ✓ | ✓ | 44.8 | 36.9 | 1,182k | 0.299 |
| | kimi-k2.5 | ✗ | ✗ | 31.2 | 18.9 | 245k | 0.070 |
| | | ✓ | ✓ | 31.2 | 20.7 | 487k | 0.103 |
| | qwen3.5-397b | ✗ | ✗ | 9.7 | 5.4 | 145k | 0.499 |
| | | ✓ | ✓ | 7.1 | 5.1 | 215k | 0.716 |

## H.2. Benchmark Result Based on Easy/Difficult Task Split

We annotate tasks into `easy` and `hard` subsets. We find that almost every model configuration performs worse on the hard subset. The only exception is `Gemini 3 Pro` which actually *improves* in the harder subset. We hypothesize this is due to the model's strong inherent multimodal understanding.

*Table 5.* `pass@1` (%) split by task difficulty. $\Delta$ is the percentage-point difference (`hard − easy`).

| Harness | Model | w/ Screenshot | w/ Video | Easy (%) | Hard (%) | $\Delta$ (pp) |
|---|---|---|---|---|---|---|
| claude-code | | ✗ | ✗ | 22.1 | 8.4 | −13.7 |
| | claude-haiku-4-5-20251001 | ✓ | ✗ | 21.4 | 11.9 | −9.5 |
| | | ✗ | ✓ | 26.0 | 13.9 | −12.1 |
| | | ✓ | ✓ | 24.4 | 11.4 | −13.0 |
| | claude-sonnet-4-5-20250929 | ✗ | ✗ | 33.6 | 25.7 | −7.8 |
| | | ✓ | ✓ | 46.6 | 27.2 | −19.3 |
| codex | | ✗ | ✗ | 51.9 | 27.2 | −24.7 |
| | gpt-5.4-mini | ✓ | ✗ | 48.9 | 30.7 | −18.2 |
| | | ✗ | ✓ | 56.5 | 34.7 | −21.8 |
| | | ✓ | ✓ | 55.0 | 28.7 | −26.2 |
| | gpt-5.4 | ✗ | ✗ | 51.1 | 34.7 | −16.5 |
| | | ✓ | ✓ | 61.1 | 46.0 | −15.0 |
| gemini-cli | | ✗ | ✗ | 50.4 | 42.1 | −8.3 |
| | gemini-3-flash-preview | ✓ | ✗ | 51.9 | 41.1 | −10.8 |
| | | ✗ | ✓ | 48.9 | 45.5 | −3.3 |
| | | ✓ | ✓ | 52.7 | 38.6 | −14.1 |
| | gemini-3-pro-preview | ✗ | ✗ | 46.6 | 52.5 | +5.9 |
| | | ✓ | ✓ | 52.7 | 54.5 | +1.8 |
| openhands | claude-haiku-4-5-20251001 | ✗ | ✗ | 23.7 | 10.4 | −13.3 |
| | | ✓ | ✓ | 31.3 | 8.9 | −22.4 |
| | gemini-3-flash-preview | ✗ | ✗ | 35.1 | 27.2 | −7.9 |
| | | ✓ | ✓ | 40.5 | 26.2 | −14.2 |
| | gpt-5.4-mini | ✗ | ✗ | 49.6 | 31.2 | −18.4 |
| | | ✓ | ✓ | 50.4 | 28.2 | −22.2 |
| | kimi-k2.5 | ✗ | ✗ | 33.6 | 9.4 | −24.2 |
| | | ✓ | ✓ | 36.6 | 10.4 | −26.2 |
| | qwen3.5-397b | ✗ | ✗ | 11.5 | 1.5 | −10.0 |
| | | ✓ | ✓ | 8.4 | 3.0 | −5.4 |

## I. Failure Analysis

We analyze failures from the four best-performing model configurations using an LLM-as-a-judge procedure. Consistent with the main paper, failures most often involve incorrect game-development patterns or failures of multimodal grounding. Since a single failed task may exhibit multiple error modes, percentages are computed over total failures and are not mutually exclusive.

*Table 6.* Common game-development failure patterns.

| Failure Type | Failures (%) |
|---|---|
| Missing or mis-parented nodes | 63.4 |
| Missing method, property, or custom signal | 36.2 |
| Unset exported reference | 35.9 |
| Incorrect physics setup | 32.1 |
| Incorrect scene or asset instantiation | 28.2 |
| Incorrect UI control-tree structure | 25.1 |
| Incorrect node type | 22.6 |
| Incorrect TileMap structure | 1.7 |

*Table 7.* Common multimodal-understanding failure patterns.

| Failure Type | Failures (%) |
|---|---|
| Incorrect shader or material assignment | 22.6 |
| Incorrect shader, post-processing, or environment parameters | 22.6 |
| Incorrect UI layout, spacing, sizing, or anchoring | 19.9 |
| Incorrect animation state, direction, or frame sequence | 17.8 |
| Incorrect camera framing, position, or view transform | 17.1 |
| Incorrect spritesheet region or atlas slice | 15.3 |
| Incorrect texture, tile, or visual asset selection | 15.3 |
| Incorrect particle emission, spread, or lifetime parameters | 15.0 |
| Collider placement inconsistent with visible sprite geometry | 2.4 |

## J. Data Contamination Analysis

To check for any data contamination, we evaluate whether models can reproduce held-out tutorial content from partial context. For each of 117 tutorial transcripts, we provide the first half of the transcript and ask the model to complete the second half. We then compare the generated continuation against the true held-out continuation using ROUGE-L and BLEU-4. A score of 1 would indicate perfect memorization. Both models obtain very low overlap scores, suggesting no evidence of systemic memorization.

*Table 8.* Transcript continuation overlap on 117 held-out tutorial transcripts.

| Model | ROUGE-L | BLEU-4 |
|---|---|---|
| Gemini 3 Flash | 0.0615 | 0.0172 |
| GPT 5.1 | 0.0623 | 0.0173 |

