# OpenReview forum: "GameDevBench: Evaluating Agentic Capabilities Through Game Development"
_ICML.cc/2026/Conference — ICML 2026 regular_

### Official Review · Reviewer_wUJR · 2026-03-05

**Soundness:** 2
**Presentation:** 3
**Significance:** 2
**Originality:** 4
**Overall Recommendation:** 4
**Confidence:** 3

**Summary:**

This paper introduces GameDevBench, a novel benchmark comprising 168 game development tasks within the Godot engine designed to evaluate the multimodal and agentic capabilities of LLMs.

**Compliance With Llm Reviewing Policy:**

Affirmed.

**Final Justification:**

The rebuttal addressed most of my concerns.

**Key Questions For Authors:**

1. See weaknesses.
2. Can you provides more specific examples to show how you constrain "all tasks can be verified through code"? To my understanding, ambiguity (same feature semantically, but different way to the ground-truth) usually occurs in game developing.

**Limitations:**

The dataset is small and static.

**Strengths And Weaknesses:**

**Strength**

Interesting and constructive. The paper successfully bridges the gap between purely text-based software engineering and multimodal computer use by introducing a context-rich, inherently multimodal game development environment.

**Weaknesses**

1. "the best agent solving only 50.0% of tasks". To my knowledge, this ratio is relatively high for a newly-proposed benchmark. The problems may be too simple?
2. At only 168 total tasks, the dataset size is prohibitively small for a modern benchmark. Given the rapid advancement of frontier models, this benchmark is at high risk of rapid saturation.
3. Lack of analysis or exploration of training a model. Better to offer a "solving" baseline.

---

> ### Author Rebuttal · Authors · 2026-03-31
>
> We thank the reviewer for their insightful feedback and hope we have addressed all of their concerns or questions.
>
> *Concern 1: "the best agent solving only 50.0% of tasks". To my knowledge, this ratio is relatively high for a newly-proposed benchmark. The problems may be too simple?*
>
> We have created a new subset of 227 harder tasks. These tasks are designed to require more multimodal understanding and target game development patterns we observed agents were deficient in. We find that the best models, even with our multimodal feedback methods, struggle on these tasks. We provide changes in performance for Sonnet 4.5 and Gemini 3 Flash below.
>
> | Model | Pass@1 (%) |
> |---|---:|
> | Claude Sonnet 4.5 | 21.4 (-23) |
> | Gemini 3 Flash | 35.6 (-14.3) |
>
> *Concern 2: At only 168 total tasks, the dataset size is prohibitively small for a modern benchmark. Given the rapid advancement of frontier models, this benchmark is at high risk of rapid saturation.*
>
> The above subset includes an additional 227 tasks bringing our total tasks to 395 tasks. This is comparable to other recent human-verified benchmarks such as SWE-Bench Verified (500 tasks), EDIT-Bench (545 tasks), or OSWorld-verified (369 tasks)
>
> *Concern 3: Lack of analysis or exploration of training a model. Better to offer a "solving" baseline.*
>
> First, we believe that the main contribution of our publication is the data and evaluation itself rather than a solution. This is increasingly common, especially as training large models becomes increasingly infeasible, and consistent with other high impact benchmark publications such as WebArena, VisualWebArena, OSWorld, or Tau-Bench.
>
> Second, we disagree with the notion that we lack analysis or exploration of improved approaches over the baseline. While we do not train a model, we do present two new methods for multimodal feedback via editor screenshots and runtime video. We show in our work that either method offers significant improvements over the baseline model. Showing improvements through new methodologies or inputs rather than training a new model is popular in other multimodal benchmarks such VisualWebArena.
>
> *Question 1 Can you provide more specific examples to show how you constrain "all tasks can be verified through code"? To my understanding, ambiguity (same feature semantically, but different way to the ground-truth) usually occurs in game development.*
>
> This is a great question. Although game development tasks may allow for many valid implementations, we can evaluate behavioral equivalence rather than exact code matching. For example, if the task is for a bullet to destroy itself and an enemy on collision, the test can spawn both objects, trigger a collision, and check that both are removed afterward. That way, regardless of how the code is implemented, as long as the behavioral requirements are met (bullet and enemy destroyed) the test will pass.

---

> > ### Author Rebuttal · Reviewer_wUJR · 2026-04-04
> >
> > Thanks for the rebuttal. Would you mind providing the detailed comparison of the two versions? (e.g. development demos in form of video/pdf/image-set, the task itself, etc.) ICML allows anonymous link under the double-blind constraint.

---

> > > ### Author Response · Authors · 2026-04-06
> > >
> > > Edit
> > > ----
> > > We're very happy to see that our analysis on the improvements in our new subset of data convinced the reviewer to increase their score. Thank you for your interest and response!
> > >
> > > Original
> > > ----
> > > Thanks for your continued interest; we really appreciate it.
> > >
> > > We’ve shared a zip file here: [https://www.dropbox.com/scl/fi/uxmr5s258c0ysznom0y96/rebuttal\_examples.zip?rlkey=5xaelqtdcrrrhr0u7xrzt7zhr\&st=sd04wtni\&dl=0](https://www.dropbox.com/scl/fi/uxmr5s258c0ysznom0y96/rebuttal_examples.zip?rlkey=5xaelqtdcrrrhr0u7xrzt7zhr&st=sd04wtni&dl=0)
> > >
> > > # Overall Summary
> > > This zip file compares the original and new subsets of data. The new tasks contain more images, lines of code, project files, and shader assets which significantly increase difficulty for models such as Gemini 3 Flash and Claude 4.5 Sonnet. Additionally, we describe our approach to creating these new tasks by increasing task context and multimodality. We also provide 2 tasks in each subset of data to compare and contrast.
> > >
> > > In the zip file, there are three folders: figures, original, and new.
> > >
> > > # Figures
> > >
> > > In the figures folder, you’ll find several images:
> > >
> > > 1. **subset\_comparison.png**. This details the comparison between the original and new subsets of data. The new tasks contain more images, lines of code, project files, and shader assets. As a result, they are on average more multimodal and more challenging compared to our original subset. As you saw in our rebuttal, performance decreased on the subset of data.
> > > 2. **data\_success.pdf**. To show that the aforementioned features are the reasons for increased difficulty, we calculated the performance differences for Gemini 3 Flash and Claude 4.5 Sonnet across different amounts of images, lines of code, project files, and shader assets. This figure shows that performance decreases as these project complexity increases.
> > > 3. **examples/\*.png or .mov.** These images or videos detail examples of various tasks in our original dataset. Tasks 54 and 132 are from the original subset. Tasks 4121 and 4123 are from the new subset. These tasks are provided in the zip.
> > >
> > > We will include all of these new results in our camera-ready.
> > >
> > > # Original Tasks
> > >
> > > The original folder contains two tasks from our original subset. These tasks are the center (task 54\) and the bottom-middle (task 0132\) images in Figure 1 of our paper. We provide both the task itself and the ground truth versions of each task.
> > >
> > > Task 54 requires adding animations to an isometric crusader figure.
> > > Task 132 requires adding a functional minimap to a top-down shooter.
> > >
> > > # New Tasks
> > >
> > > The new folder contains two tasks from our new subset also with both the task and its ground truth provided.
> > >
> > > Task 4121 requires adding animations and functionality for punching (press c in the ground truth to punch; it’s a playable game).
> > > Task 4123 requires separating out pipe tiles into its own layer in a tilemap.
> > >
> > > # Task Comparison
> > >
> > > We learned two lessons that we applied to the construction of our new subset of tasks.
> > >
> > > 1\. **Tasks benefit from additional context.** We learned that the original generation process stripped out much of the surrounding context of a task. See Task 54, where the isometric crusader is isolated without any surrounding game context. The task is still complex and challenging, but we thought it could be improved with more context. See Task 4121 where the animations and actions are now in the context of the game and fully playable.
> > > 2\. **Multimodal tasks are challenging.** One main finding from our experiments was that models struggle on multimodal tasks. As a result, tasks in the new subset focus on multimodal capabilities, requiring that agents observe the scene to solve the task. For example, in Task 4123, the agent must conceptually understand what tiles are “pipe” tiles in order to create the appropriate layer mask. In Task 4121, the agent must understand which portions of the spritesheet are related to the “punch” animation.
> > >
> > > # Extra Information
> > >
> > > If the reviewer chooses to open the files please note that you must first open it in the editor via:
> > >
> > > ```godot --editor --path /path/to/folder```
> > >
> > > To allow the game assets to load. You can press the play button on the top right to play the scene. Alternatively execute (after loading in the editor) the following command to play the scene:
> > >
> > > ```godot --path /path/to/folder```

---

### Official Review · Reviewer_zQmR · 2026-03-06

**Soundness:** 3
**Presentation:** 3
**Significance:** 3
**Originality:** 3
**Overall Recommendation:** 4
**Confidence:** 3

**Summary:**

This paper introduces GameDevBench, a benchmark designed to evaluate the agentic capabilities of models on game development tasks. The benchmark construction pipeline includes several stages: data preparation from videos and web sources, automatic task construction using a Codex-based agent, task refinement, and task categorization using GPT-5-mini.
The benchmark includes diverse tasks involving multiple file types and multimodal inputs. The authors evaluate several models on the proposed benchmark and analyze the results, including the impact of multimodal information such as runtime video instructions and the use of MCP.

**Compliance With Llm Reviewing Policy:**

Affirmed.

**Final Justification:**

I will maintain my score. While the paper is solid, it does not particularly stand out.

**Key Questions For Authors:**

Could the tasks be further refined through more fine-grained categorization or difficulty grading to enable a more detailed analysis of model capabilities and performance?

**Limitations:**

yes

**Strengths And Weaknesses:**

Strengths

1. Diverse and multimodal benchmark design.
The proposed benchmark spans a variety of task types and file formats and naturally incorporates multimodal inputs. The tasks are also designed to allow deterministic verification, which improves evaluation reliability.

2. Evaluation of multimodal assistance and tool usage.
The paper compares model performance with and without MCP and runtime video instructions. The results show that providing multimodal information significantly improves performance.

3. Analysis across task categories.
By analyzing model performance across different task categories, the paper highlights that improvements in multimodal understanding are particularly important for game development tasks.

Weaknesses

1. Coarse task categorization.
The current task categorization appears relatively coarse. A more fine-grained taxonomy could enable a more detailed analysis of the specific capabilities and limitations of different models.

2. Lack of task difficulty grading.
The benchmark would benefit from grading tasks by difficulty or complexity. This would make it easier to analyze model performance across different levels of challenge and better understand the capability boundaries of current models.

3. Limited interaction setting for agents.
The evaluation primarily focuses on code generation. Allowing agents to interact directly with development environments (e.g., editors) could provide a more comprehensive evaluation of real-world agent capabilities in game development workflows.

---

> ### Author Rebuttal · Authors · 2026-03-31
>
> We thank the reviewer for their feedback and hope we have addressed all of their concerns or questions.
>
> *Concern 1: Coarse task categorization. The current task categorization appears relatively coarse. A more fine-grained taxonomy could enable a more detailed analysis of the specific capabilities and limitations of different models.*
>
> We were unable to find a comprehensive and fine-grained game development taxonomy in the literature. We agree a comprehensive taxonomy is important and warrants its own publication.
>
> *Concern 2: Lack of task difficulty grading. The benchmark would benefit from grading tasks by difficulty or complexity. This would make it easier to analyze model performance across different levels of challenge and better understand the capability boundaries of current models.*
>
> We are tackling task difficulty in two ways. First, we have created a new subset of 227 harder tasks. These tasks are designed to require more multimodal understanding and target game development patterns we observed agents were deficient in. We find that the best models, even with our multimodal feedback methods, struggle on these tasks. We provide changes in performance for Sonnet 4.5 and Gemini 3 Flash below.
>
> | Model | Pass@1 (%) |
> |---|---:|
> | Claude Sonnet 4.5 | 21.4 (-23) |
> | Gemini 3 Flash | 35.6 (-14.3) |
>
> Second, we will have our annotators go back and grade all tasks. We will then see how well those difficult labels calibrate with model performance. By doing so, we will be able to directly tackle your suggestions and improve our understanding of model capabilities for game development.
>
> *Concern 3: Limited interaction setting for agents. The evaluation primarily focuses on code generation. Allowing agents to interact directly with development environments (e.g., editors) could provide a more comprehensive evaluation of real-world agent capabilities in game development workflows.*
>
> Currently, computer use agents (CUAs) perform worse than their software counterparts (see [Aggrawal, 2025](https://arxiv.org/abs/2502.18525)). Ultimately, we believe that being able to directly modify code in Godot is one of its strengths of the game engine w.r.t. benchmarking multimodal agents, and chose to focus on that for this work. GameDevBench is designed such that any future CUA implementation can be directly tested on the tasks in the same way as the current approaches.

---

> > ### Author Rebuttal · Reviewer_zQmR · 2026-04-03
> >
> > Thanks

---

> > > ### Author Response · Authors · 2026-04-05
> > >
> > > Thanks for the response and we're happy to hear it resolved your concerns!
> > > Since we resolved your concerns, if you would consider increasing your positive score further we would appreciate it greatly. Thank you.

---

### Official Review · Reviewer_zLPM · 2026-03-12

**Soundness:** 3
**Presentation:** 3
**Significance:** 3
**Originality:** 3
**Overall Recommendation:** 5
**Confidence:** 3

**Summary:**

This paper introduces GameDevBench, a benchmark for evaluating AI agents' capabilities in game development tasks within the Godot engine. Game development is a good setting for multimodal agents because it combines large codebases with visual and interactive components such as images, animations, and scene structures. To construct the benchmark, the authors generate tasks from publicly available game development tutorials and repositories, combining automated task generation with human refinement to ensure task quality. The resulting benchmark contains 168 tasks spanning multiple categories such as gameplay logic, graphics, and user interface development, each with deterministic tests for automatic evaluation.

The authors evaluate several language models across different agent frameworks and find that current systems struggle with these tasks, achieving only moderate success rates. Performance drops substantially for smaller models and for tasks requiring stronger multimodal reasoning. The study also explores simple multimodal feedback mechanisms, such as providing editor screenshots or runtime video, which lead to noticeable performance improvements. An error analysis highlights two common failure modes: limited multimodal understanding (e.g., selecting the wrong visual assets) and difficulty handling common game development patterns.

**Compliance With Llm Reviewing Policy:**

Affirmed.

**Final Justification:**

The rebuttal addressed my main concerns.

**Key Questions For Authors:**

1. **Task generation bias:** How does the choice of tutorials and the use of an automated agent for task generation influence the diversity and difficulty of the benchmark tasks? Would a purely human-curated task set produce a significantly different distribution of skills or complexity?

2. **Understanding multimodal tool benefits:** The paper demonstrates that screenshot and video feedback can improve performance. Have the authors analyzed agent trajectories or logs to determine how models use this additional information (e.g., debugging, verification, or exploration)?

3. **Task quality after filtering:** The paper mentions that a small percentage of automatically generated tasks were unfixable and removed. Could the authors elaborate on the types of issues encountered and whether they reveal systematic limitations of the task generation pipeline?

**Limitations:**

Yes

**Strengths And Weaknesses:**

### Soundness

**Strengths**
- The paper presents a clear multi-stage pipeline for constructing tasks from tutorials, combining automated generation with human refinement and final annotation to ensure task quality.
- Using Godot’s testing framework enables deterministic verification through code execution instead of subjective evaluation methods.
- The manual error analysis provides useful insights into common failure modes such as multimodal perception and game-development patterns.

**Weaknesses**
- Many tasks are generated using a Codex agent from tutorial transcripts and repositories, which may introduce biases or artifacts despite human refinement.
- Evaluation focuses mainly on pass@1 success rates, offering only a binary measure of completion. Metrics such as pass@k, time-to-solution, or partial test coverage would provide a more nuanced evaluation.
- Because tasks are derived from tutorials, the benchmark may overrepresent common development patterns while underrepresenting more complex real-world workflows.

---

### Presentation

**Strengths**
- The paper is well structured and clearly explains the dataset construction pipeline.
- Figures illustrating the Godot editor environment and task structure help convey the multimodal nature of the benchmark.

**Weaknesses**
- Some result tables are dense and could benefit from clearer summarization or visualization.
- The paper reports pass@1 as the main metric but does not explicitly define it, which may reduce clarity for readers unfamiliar with the convention.

---

### Significance

**Strengths**
- Game development provides a strong testbed for multimodal agents, combining code generation, visual reasoning, and structured scene representations.
- Unlike existing coding benchmarks that focus on unimodal settings, this benchmark evaluates tasks requiring both code and visual understanding.
- The benchmark may encourage research on agents that operate in complex software environments with multimodal inputs.

**Weaknesses**
- Most tasks focus on modifying existing projects rather than creating systems from scratch, which may limit the diversity of development challenges.

---

### Originality

**Strengths**
- The work introduces one of the first benchmarks specifically targeting agent performance in game-development environments.
- The use of screenshot and video feedback mechanisms provides a practical approach to improving agent interaction with the environment.

---

> ### Author Rebuttal · Authors · 2026-03-30
>
> We thank the reviewer for their positive feedback and hope we have addressed all of their concerns or questions.
>
> *Concern 1: Evaluation focuses mainly on pass@1 success rates, offering only a binary measure of completion. Metrics such as pass@k, time-to-solution, or partial test coverage would provide a more nuanced evaluation.*
>
> We have conducted additional analysis on cost-per-solution. We share some of our highlights and results here. We will include these results in our camera-ready submission.
>
> First, Gemini 3 Flash is the most cost-effective model, achieving best performance with best cost. Second, model framework dramatically affects cost-effectiveness. For example, using Sonnet 4.5 costs 15x more in OpenHands compared to in Claude Code. Third, model capacity and per-token cost does not necessarily correlate with final task cost. For example, when using Claude Code, Claude Opus 4.5 costs half as much as Claude Sonnet 4.5 despite offering improved performance
>
>  | Model | Baseline Pass@1 (%) | Baseline Cost ($) | Multimodal Pass@1 (%) | Multimodal Cost ($) |
>   |---|---:|---:|---:|---:|
>   | Claude Sonnet 4.5 | 33.33 | 0.04 | 46.21 | 0.42 |
>   | Claude Opus 4.5 | 39.39 | 0.09 | 50.00 | 0.23 |
>   | Gemini 3 Flash | 46.97 | 0.13 | 52.27 | 0.20 |
>
>   | Model | Framework | Pass@1 (%) | Cost ($) |
>   |---|---|---:|---:|
>   | Claude Sonnet 4.5 | claude-code | 33.33 | 0.04 |
>   | Claude Sonnet 4.5 | openhands | 43.18 | 0.60 |
>   | GPT-5.1 Codex Max | codex | 43.59 | 0.20 |
>   | GPT-5.1 Codex Max | openhands | 45.45 | 0.38 |
>
> *Concern 2: Tasks are derived from tutorials which may limit task difficulty, task diversity, and introduce biases.*
>
> These tasks are not meant to mirror complex real-world workflows, but instead focus on medium complexity tutorials. Even then, models struggle with many of the tasks in the benchmark. We see this benchmark as a starting point for agentic game development, not an end point and hope to see future work in this direction.
>
> *Concern 3: Some result tables are dense and could benefit from clearer summarization or visualization.*
>
> Assuming the reviewer is referring to our main results table (Table 2), we will complement this table with a scatterplot visualization of cost vs. pass@1 success rate, better highlighting the Pareto frontier of models. If the reviewer has another suggestion, we would be more than happy to improve or amend them.
>
> *Concern 4: The paper reports pass@1 as the main metric but does not explicitly define it, which may reduce clarity for readers unfamiliar with the convention.*
>
> We will add a definition in our paper.
>
> *Question 1: How would a human-curated task set differ from tutorials in terms of skills or complexity?*
>
> We attempted to create a purely human-curated task set at the onset of this project. A few key challenges and differences. First, humans tended to create much more multimodal-oriented tasks. Second, it was difficult to create long and complex tasks. This is sensible as tutorial construction (i.e., making a YouTube tutorial) is a time-consuming process for humans. Ultimately, we opted for the tutorial-based task derivation pipeline since it was more scalable.
>
> *Question 2: Have the authors analyzed how agents utilize screenshot and video feedback?*
>
> We have inspected several agent trajectories. We have found that models use screenshot and video feedback exactly as expected, checking visually to see if the visual state aligns with their expectation. As noted in the paper, one interesting feature is that models tend to prefer screenshots, either capturing only 1 frame or turning video into frames via Python scripts.
>
> *Question 3: What issues were encountered and did they reveal any systematic issues with the pipeline?*
>
> There were a few main issues. First, since models struggle with multimodal understanding, they can sometimes struggle with creating sensible multimodal tasks as well. Oftentimes these are little details such as camera positioning (which we note in our paper) and can be fixed, but occasionally they are unfixable. Second, occasionally models create tasks that are impossible to solve without watching the tutorial video itself. We resolve this through prompt engineering, but occasionally the model still fails. Lastly, some projects were simply broken to begin with and undetectable until after the task was created. Note that through extensive prompt engineering and a separate validation step (as mentioned in our pipeline), we were able to significantly reduce the frequency of these issues.

---

> > ### Author Rebuttal · Reviewer_zLPM · 2026-04-04
> >
> > Thanks for the author's response, and it resolved my concerns. I'll keep my current score.

---

> > > ### Author Response · Authors · 2026-04-05
> > >
> > > Thank you for the response and we're happy to hear it resolved your concerns!

---

### Official Review · Reviewer_63ao · 2026-03-13

**Soundness:** 2
**Presentation:** 3
**Significance:** 3
**Originality:** 3
**Overall Recommendation:** 4
**Confidence:** 3

**Summary:**

This paper introduces GameDevBench, a benchmark of 168 game development tasks in the Godot engine, derived from web and video tutorials. The benchmark tests multimodal agent capabilities, tasks require understanding images, sprites, 3D assets, and game engine patterns. The authors evaluate multiple frontier models (Claude, Gemini, GPT, Kimi, Qwen) across different agentic frameworks (claude-code, gemini-cli, codex, OpenHands), and introduce two visual feedback methods: an MCP server for editor screenshots and runtime video generation. Best agent solves 50% of tasks, visual feedback nearly doubles performance in one setting, and performance drops sharply on graphics-heavy tasks.

**Compliance With Llm Reviewing Policy:**

Affirmed.

**Final Justification:**

My concerns have been solved.

**Key Questions For Authors:**

1. How many tasks were discarded during construction?
2. Have you measured how long a competent Godot developer takes on these tasks, and what their pass@1 would be?
3. Godot tutorials are public. Have you checked whether the models were trained on these tutorials? A simple contamination check (e.g., prompting models to complete tutorial text) would strengthen the paper.

**Limitations:**

Yes.

**Strengths And Weaknesses:**

**Strengths:**

- S1: Game development is a genuinely compelling testbed for multimodal agents. The paper correctly identifies that it bridges the gap between unimodal code generation and full multimodal computer use. You still write code, but you need to understand sprites, animations, scene trees, and visual outputs. This is a real gap in the benchmark landscape.

- S2: The MCP editor screenshot tool and runtime video feedback are simple but effective interventions. The finding that Claude Sonnet 4.5 nearly doubles from 25.6% to 44.4% with video feedback (Table 2) is striking and practically useful. This is the kind of result that practitioners can immediately act on.

- S3: Testing the same model across different agentic frameworks (e.g., Claude Sonnet 4.5: 25.6% in claude-code vs 38.9% in OpenHands) reveals that framework choice matters as much as model choice in some cases. This is an underexplored dimension in most agent benchmarks.

**Weaknesses:**

- W1: 168 tasks, with results often reported on subsets (Table 2 appears to use 90 tasks based on percentages). For a benchmark paper at ICML, this is small. The per-category breakdowns must have very few tasks each, making the skill-level analysis in Figure 5 statistically noisy.

- W2: Tasks are constructed semi-automatically by an LLM agent (Appendix A shows the full Codex prompt for converting YouTube tutorials to tasks). While human verification is mentioned, the paper doesn't quantify how many tasks were discarded or edited during quality control. Given the complexity of Godot projects, how many tasks have incorrect ground truth? The paper doesn't report any inter-annotator agreement or error analysis on the tasks themselves.

- W3: The error analysis (Section 4.3) is qualitative and brief. Just two paragraphs identifying "multimodal understanding" and "game development patterns" as failure modes. No quantitative breakdown of error types, no failure case statistics. Compare to the more rigorous error taxonomies in SWE-bench Multimodal.

- W4: The paper uses only pass@1. This is fine but coarse, a task that is 90% correct gets the same score as one that doesn't run. Partial credit metrics or a more granular scoring scheme would be more informative, especially for game development where "mostly works but animation is wrong" is a common outcome.

- W5: No human baseline is reported. How well do junior/senior game developers do on these tasks? Without this, it's hard to calibrate the 50% solve rate. Is this impressive or terrible?

---

> ### Author Rebuttal · Authors · 2026-03-30
>
> We thank the reviewer for their feedback and hope to have addressed their concerns.
>
> *Concern 1: For a benchmark paper at ICML, the task amount is small*
>
> As mentioned in Section 3.2, GameDevBench’s pipeline is repeatable with more tutorial sources. Since submission, we have gathered a new subset of 227 tasks, bringing our total tasks to 395 tasks. This is comparable to other recent human-verified benchmarks such as SWE-Bench Verified (500 tasks), EDIT-Bench (545 tasks), or OSWorld-verified (369 tasks). We will add these tasks to our camera-ready.
>
> *Concern 2: How many tasks were discarded or edited during quality control? How many tasks have incorrect ground truth?*
>
> In Section 2.3, we discuss our study on task generation where we found that 93% of tasks were either acceptable or only required minor modifications. 7% of tasks were discarded due to infeasibility. We found this study held through our annotation pipeline. We later pruned tasks that were too simple to be included (e.g., tasks that required single node or single file additions). We will include this detail in our camera-ready.
>
> *Concern 3: Can there be a more rigorous error taxonomy similar to that found in SWE-Bench Multimodal?*
>
> We have conducted a more thorough analysis of the failure modes. Using an LLM-as-a-Judge approach, we analyzed all failures for the 4 best-performing models. As noted in our original analysis, failures were most commonly associated with game development patterns and multimodal understanding. We further broke down the failures into specific patterns (each task can have multiple failure patterns).
>
>   **Game-Development Failures**
>   | Failure Type | % of Total Failures |
>   |---|---:|
>   | Missing or mis-parented nodes | 63.4 |
>   | Missing method, property, or custom signal | 36.2 |
>   | Unset exported reference | 35.9 |
>   | Incorrect physics setup | 32.1 |
>   | Incorrect scene or asset instantiation | 28.2 |
>   | Incorrect UI control-tree structure | 25.1 |
>   | Incorrect node type | 22.6 |
>   | Incorrect TileMap structure | 1.7 |
>
>   **Multimodal Understanding Failures**
>   | Failure Type| % of Total Failures |
>   |---|---:|
>   | Incorrect shader or material assignment | 22.6 |
>   | Incorrect shader, post-processing, or environment parameters | 22.6 |
>   | Incorrect UI layout, spacing, sizing, or anchoring | 19.9 |
>   | Incorrect animation state, direction, or frame sequence | 17.8 |
>   | Incorrect camera framing, position, or view transform | 17.1 |
>   | Incorrect spritesheet region or atlas slice | 15.3 |
>   | Incorrect texture, tile, or visual asset selection | 15.3 |
>   | Incorrect particle emission, spread, or lifetime parameters | 15.0 |
>   | Collider placement inconsistent with visible sprite geometry | 2.4 |
>
> Compared to SWE-Bench Multimodal, which only has 4 failure patterns, these results emphasize the diversity of tasks and failure patterns present in GameDevBench. We will include these results in our camera-ready.
>
> *Concern 4: The paper uses only pass@1 and would benefit from more other analysis on results.*
>
> For additional analysis, we have analyzed cost-per-solution. We share some highlighted results here. We will include these in our camera-ready submission.
>
> First, Gemini 3 Flash is the most cost-effective model, achieving best performance with best cost. Second, model framework dramatically affects cost-effectiveness. For example, Sonnet 4.5 costs 15x more in OpenHands compared to in Claude Code. Third, model capacity and per-token cost does not necessarily correlate with final cost. For example, when using Claude Code, Claude Opus 4.5 costs half as much as Claude Sonnet 4.5 despite offering improved performance.
>
> Due to character limits, please see the table in our response to reviewer Reviewer zLPM under Concern 1.
>
> *Concern 5: No human baseline is reported. How well do game developers do on these tasks?*
>
> Since tasks are sourced from game development tutorials, they are not challenging for the average game developer and intended as first steps towards game development. Annotators were able to solve all tasks. Time to completion among humans varies as many tasks require understanding multiple project files or scenes. Simple multimodal tasks take around 2 minutes while longer, code-oriented tasks can take up to 30 minutes. Note that this is an inverse trend when compared to agents, where multimodal tasks are more challenging than code-oriented tasks.
>
> *Question: Is there any data contamination in the benchmark?*
>
> Great question. We checked to see if models could complete the second half of the tutorial transcripts given the first half. We evaluated 117 transcripts and found no evidence of systemic memorization (a score of 1 indicates perfect memorization). We will include this in our camera-ready.
>
> | Model | ROUGE-L | BLEU-4 |
>   |---|---|---|
>   | Gemini 3 Flash | 0.0615 | 0.0172 |
>   | GPT 5.1 | 0.0623 | 0.0173 |

---

> > ### Author Rebuttal · Reviewer_63ao · 2026-04-07
> >
> > My concerns have been solved. Thank you for the rebuttal.

---

> > > ### Author Response · Authors · 2026-04-08
> > >
> > > We're very happy to see that our expanded dataset and in-depth analysis convinced the reviewer to improve their score. Thank you for the response!

---

### Decision · Program_Chairs · 2026-04-30

**Decision:**

Accept (regular)

**Comment:**

GameDevBench introduces 168 game development tasks in the Godot engine to evaluate multimodal agentic capabilities. Tasks combine code generation with visual understanding, and the paper evaluates frontier models across agent frameworks, introduces screenshot/video feedback methods that nearly double performance in some settings, and shows sharp drops on graphics-heavy tasks.

Reviewers agreed on the benchmark's novelty and practical value. Initial concerns around dataset size, task difficulty, coarse categorization, human baselines, and contamination were largely addressed in the rebuttal: the dataset was expanded to 395 tasks with a harder subset where top models drop substantially (Sonnet 4.5 to 21.4%), a detailed failure taxonomy was added, and contamination checks showed no systematic memorization.